

# Stop signals delay synchrony more for finger tapping than vocalization: a dual modality study of rhythmic synchronization in the stop signal task

Leidy J. Castro-Meneses[1,2,3,*] and Paul F. Sowman[1,2,*]

[1] Perception in Action Research Centre (PARC), Department of Cognitive Science, Macquarie University, North Ryde, NSW, Australia
[2] Australian Research Council Centre of Excellence in Cognition and its Disorders (CCD), Macquarie University, North Ryde, NSW, Australia
[3] The MARCS Institute for Brain, Behaviour and Development, University of Western Sydney, Bankstown, NSW, Australia
* These authors contributed equally to this work.

Corresponding author
Leidy J. Castro-Meneses,
l.castro-meneses
@westernsydney.edu.au

## ABSTRACT

**Background:** A robust feature of sensorimotor synchronization (SMS) performance in finger tapping to an auditory pacing signal is the negative asynchrony of the tap with respect to the pacing signal. The Paillard–Fraisse hypothesis suggests that negative asynchrony is a result of inter-modal integration, in which the brain compares sensory information across two modalities (auditory and tactile). The current study compared the asynchronies of vocalizations and finger tapping in time to an auditory pacing signal. Our first hypothesis was that vocalizations have less negative asynchrony compared to finger tapping due to the requirement for sensory integration within only a single (auditory) modality (intra-modal integration). However, due to the different measurements for vocalizations and finger responses, interpreting the comparison between these two response modalities is problematic. To address this problem, we included stop signals in the synchronization task. The rationale for this manipulation was that stop signals would perturb synchronization more in the inter-modal compared to the intra-modal task. We hypothesized that the inclusion of stop signals induce proactive inhibition, which reduces negative asynchrony. We further hypothesized that any reduction in negative asynchrony occurs to a lesser degree for vocalization than for finger tapping.

**Method:** A total of 30 participants took part in this study. We compared SMS in a single sensory modality (vocalizations (or auditory) to auditory pacing signal) to a dual sensory modality (fingers (or tactile) to auditory pacing signal). The task was combined with a stop signal task in which stop signals were relevant in some blocks and irrelevant in others. Response-to-pacing signal asynchronies and stop signal reaction times were compared across modalities and across the two types of stop signal blocks.

**Results:** In the blocks where stopping was irrelevant, we found that vocalization (−61.47 ms) was more synchronous with the auditory pacing signal compared to finger tapping (−128.29 ms). In the blocks where stopping was relevant, stop signals induced proactive inhibition, shifting the response times later. However, proactive

inhibition (26.11 ms) was less evident for vocalizations compared to finger tapping (58.06 ms).

**Discussion:** These results support the interpretation that relatively large negative asynchrony in finger tapping is a consequence of inter-modal integration, whereas smaller asynchrony is associated with intra-modal integration. This study also supports the interpretation that intra-modal integration is more sensitive to synchronization discrepancies compared to inter-modal integration.

## INTRODUCTION

The sensorimotor synchronization (SMS) task is used to study the ability of humans to coordinate their movements in time with an external beat. The most common experimental conceptualization of this task is finger tapping, whereby participants tap with a finger in time to an auditory pacing stimulus (*Repp, 2005*). Performance on this task is a useful index of rhythmic timing ability and has been used to investigate, amongst other things, disorders in which timing is thought to be impaired (*Carroll et al., 2009*; *Falk, Müller & Dalla Bella, 2015*; *Rubia et al., 1999*), the plasticity of timing that occurs with musical training (*Aschersleben, 2002*; *Repp, 2005*) and to study multiple timing mechanisms (*Iversen & Balasubramaniam, 2016*; *Studenka, Zelaznik & Balasubramaniam, 2012*). The finger-tapping task gives rise to two main metrics that index rhythmic timing performance. The first relates to the accuracy of tapping in relation to the timing of the pacing rhythm and is usually termed the "asynchrony". The second measure obtained from this task is the variability of inter-tap intervals, which refers to how regular taps are, relative to the pacing stimulus (*Chen, Ding & Scott Kelso, 2001*; *Iversen & Balasubramaniam, 2016*; *Studenka, Zelaznik & Balasubramaniam, 2012*).

An interesting phenomenon that is observed during finger tapping to auditory pacing signal is that humans tap, not at the point of absolute temporal coincidence between tap and auditory pacing stimulus, but at about 20–100 ms prior to the auditory pacing signal (see *Aschersleben, 2002*; *Sugano, Keetels & Vroomen, 2012*). This anticipatory relationship is referred to as "negative asynchrony", first described more than 100 years ago (*Dunlap, 1910*; *Johnson, 1899*; *Wallin, 1904*). The mechanism that gives rise to this synchronization error is not agreed upon, and may well turn out to be multi-factorial.

There are a number of explanations for the negative mean asynchrony (NMA). These explanations may be competing but not mutually exclusive (see *Aschersleben, 2002*; *Repp, 2005*; *Repp & Su, 2013* for reviews). The transmission hypothesis, also called the "Paillard–Fraisse" hypothesis, proposed by Fraisse in 1980 and who credited it to Paillard (cited in *Aschersleben & Prinz, 1995*) argues that the NMA comes for different conduction times between a sensory channel and the brain. Another explanation of the NMA is the sensory accumulator hypothesis (*Aschersleben, 2002*) that argues that the

NMA is a result of a central representation of the average of various sensory channels processed at different rates. Unlike a peripheral coding of sensory information in the Paillard–Fraisse hypothesis, the sensory accumulator hypothesis proposes an overall central coding of sensory information.

There are other theories that explain the NMA from a biomechanical perspective. For example, the optimal selection hypothesis (*Vaughan et al., 1996*) argues that in finger tapping, the energy used is minimized while satisfying overt task demands by simultaneously selecting one or two optimal limb resonant frequencies that match the task (driving) frequency. Unlike the two previous hypotheses that explained NMA as a function of the transmission times of sensory channels, the optimal selection hypothesis explains NMA as a function of the optimal effector's resonant frequency. See also *Pikovsky, Rosenblum & Kurths (2002)* that give a more holistic overview of how tapping can be regarded as an oscillatory motor activity that is coupled to an external driving oscillator (i.e., the pacing signal. See *Strogatz, 2003* for a book review). In a similar fashion, *Hove et al. (2013)* found that certain pacing stimuli produce better synchronization than others in finger tapping. Specifically, beeps are more accurately synchronized to than continuous sirens and that moving stimuli are more accurately synchronized to than static flashes. The explanation of these phenomena begets the "modality appropriateness" theory, which suggests that auditory and visual systems may have a greater capacity for encoding certain stimuli than others.

Another explanation for the NMA is the linear phase correction theory (*Schulze & Vorberg, 2002*) that explains synchronization via a statistical argument in which participants try to minimize variance of their asynchronies through a linear phase correction. See also *Large & Jones (1999)* for a description of other statistical theories.

In a slightly different proposal of how synchronization occur, *Large & Jones (1999)* suggest the "attentional dynamics" theory. This theory postulates two entities: external rhythms, which are created by the outside events, and internal rhythms, which are generated by temporal expectancies. It postulates that external and internal rhythms are coordinated via entrainment.

The present study presents evidence in favor of the Paillard–Fraisee hypothesis. Thus, in the following we explain this account in more detail. This hypothesis suggests that sensory transduction and afferent conduction times play a crucial role in SMS. Sensory transduction refers to the conversion, by a sensory receptor, of a physical stimulus into an electrical potential that is then able to be transmitted to the central nervous system (*Lumpkin & Caterina, 2007*). Afferent conduction time, on the other hand, refers to the time that elapses as information is carried from a sensory neuron to the central nervous system.

In the context of finger tapping to an auditory pacing signal, the Paillard–Fraisee hypothesis explains that the NMA occurs because the brain is synchronizing to two sensory modalities: the tactile/kinaesthetic sensory modality and the auditory modality. Given that the tactile/kinaesthetic modality takes a longer time to reach the brain due to differences in sensory transduction and afferent conduction times compared to the auditory sensory modality, coincident stimulation of both sensory channels would give

rise to asynchronous registration in the brain. The consequence of this is that tapping to an auditory pacing signal can be perceived to be "in time", and not ahead of the beat as is the veridical case.

Example evidence supporting the Paillard–Fraisee hypothesis comes from an experiment where participants were informed of the size and direction of their asynchronies, they were subsequently able to tap with greatly reduced NMA, reporting a subjective experience of having to delay their responses (*Aschersleben, 2002*, pp. 67–68). In another study, when participants received a delayed feedback after their taps, the asynchronies became more negative than for a no-delayed feedback condition (*Sugano, Keetels & Vroomen, 2012*). This finding indicates that participants tried to recalibrate their taps to their perception of temporal coincidence. This temporal recalibration has also been observed in delayed visual feedback (*Cunningham, Billock & Tsou, 2001*; *Stetson et al., 2006*) and tactile motor recalibration (*Heron, Hanson & Whitaker, 2009*).

More evidence supporting the Paillard–Fraisee hypothesis comes from an experiment showing that foot tapping, as opposed to finger tapping, to an auditory pacing signal exhibits a larger negative asynchrony, possibly because the tactile afferent conduction time of the foot is longer compared to that of the hand (*Aschersleben & Prinz, 1995*; *Billon et al., 1996*).

*Müller et al. (2008)* compared tactile and auditory pacing signals in finger and toe tapping tasks. They found that negative asynchrony was significantly smaller for both finger and toe tapping when the pacing signal was tactile compared to auditory. While finger and toe tapping to a tactile pacing signal had a NMA of approximately 8 ms (not statistically different from the onset of the tactile pacing signal), finger and toe tapping to an auditory pacing signal exhibited a NMA of approximately 40 ms that was significantly negative with reference to the onset of the auditory pacing signal. The authors interpreted this finding as evidence to support the argument that SMS is affected by whether sensory integration is inter- or intra-modal. Because the sensory feedback generated by finger and toe tapping occur within the same sensory modality as the tactile pacing signal (i.e., intra-modal integration), the signals required for synchronization are more temporally coincident centrally (in the brain). In contrast, finger and toe tapping to an auditory pacing signal requires integration across two sensory modalities (i.e., inter-modal integration) that have different sensory transduction and conduction times. This difference causes a temporal offset centrally that results in a large NMA.

It remains an open question as to whether the findings introduced above describe a general rule regarding the synchronization consequences determined by intra- versus inter-modal sensory integration contexts, as to date, the descriptions of this phenomenon are limited to the synchronization of effectors controlled by the corticospinal system (like the finger or toe) to a tactile or auditory pacing signals. Therefore, in order to advance this line of research we developed the current study to test SMS in another sensory/effector modality pairing, that being an auditory pacing signal with a vocal response and compared this to the classic finger tapping to an auditory pacing signal task. Finger tapping with auditory pacing requires the tactile/kinaesthetic and auditory sensory

modalities respectively. The tactile/kinaesthetic modality refers to finger tapping, while the auditory modality refers to the auditory pacing signal.

On the other hand, the sensory integration required for vocalization to an auditory pacing signal occurs within the same sensory modality because we assume that vocalization produces a sound, which is equivalent to the auditory feedback of experiment 2 in *Aschersleben & Prinz (1995)*. In experiment 2 of *Aschersleben & Prinz (1995)*, finger taps elicited auditory feedback. Because the auditory feedback occurred within the same sensory modality as the auditory pacing signal, the authors interpreted that the relative phase of finger tapping would be modulated by both inter and intra-modal integration processes. *Aschersleben & Prinz (1995)* found that finger tapping with auditory feedback exhibited a relative reduction in NMA of 17 ms compared to finger tapping without auditory feedback.

According to this previous study (*Aschersleben & Prinz, 1995*), we predicted that finger tapping (with no auditory feedback) to an auditory pacing signal would show a larger NMA due to the requirement for inter-modal integration (i.e., differences in sensory transduction and afferent conduction times). In contrast to finger tapping, in the intra-modal case, vocalizations to an auditory pacing stimulus would show significantly less NMA, in accordance with *Müller et al. (2008)* and *Aschersleben & Prinz (1995)*.

However, a significant problem arises when attempting to test this hypothesis. Finger tapping and vocalizations are measured via two different methods. To calibrate the time constants between these two measures is unlikely to be precise. Our proposed solution to this problem was to measure and compare the responses to a perturbation of the synchronization time across response modalities. We based our perturbation on that presented in *Fischer et al. (2016)* who demonstrated that synchronization time is modulated in the presence of a secondary inhibitory control task: the stop signal task (*Logan & Cowan, 1984*).

The stop signal task is a paradigm that measures the time it takes to stop an ongoing response. The experimental task consists of go and stop trials. In go trials, participants respond rapidly to a go signal. Stop trials start out as go trials but following the go signal, a stop signal appears. The stop signal indicates that participants should attempt to arrest their already initiated response. Stop trials usually make up a minority of all trials (e.g., 25% is a typical proportion of stop trials). The speed at which participants are able to stop is indexed by the stop signal reaction time (SSRT). SSRT is considered primarily a measure of reactive inhibition because participants react to the stop signals by attempting to arrest their response (*Aron, 2011*).

Another measure that can be obtained from variants of the stop signal task is proactive inhibition (*Chikazoe et al., 2009*; *Jaffard et al., 2008*). This refers to how much participants prepare to stop in anticipation of stop signals by slowing their go responses. To measure proactive inhibition, the experimental task requires a control condition in which stopping is not required. The go reaction time (go RT) in the control condition is then compared to the go RT in the condition where stopping is required (*Castro-Meneses, Johnson & Sowman, 2015*; *Castro-Meneses, Johnson & Sowman, 2016*; *Chikazoe et al., 2009*; *Jaffard et al., 2008*). Generally, participants exhibit slower go RTs under

conditions where stopping is required, which is thought to be indicative of advanced preparation for possible stop signals (*Duque et al., 2017*), or proactive inhibition.

*Castro-Meneses, Johnson & Sowman (2016)* used two types of blocks to measure proactive inhibition. In their study, some blocks contained go and stop trials; this was the relevant stop condition. Other blocks contained go and stop trials but participants were instructed to ignore the stop trials and treat them as if they were go trials. These blocks were the irrelevant stop condition. The irrelevant stop blocks served as the control condition, which go RT changes could be assessed against the relevant stop blocks (i.e., when stopping was required). It was found that when stopping was required in the relevant stop condition, participants slowed their go RTs compared to those RTs in the irrelevant stop condition.

The current experiment utilizes a combination of the SST and the SMS as per *Fischer et al. (2016)*. In accordance with the results of their study, we predicted that the anticipation of a stop signals before the pacing signal would induce proactive inhibition, delaying the synchronization response. This effect would be indexed by a reduction in the NMA. Although, slower go RTs in the stop signal task are considered to represent advanced preparation to stop (or proactive inhibition), we interpret that in a SMS with visual stop signals, a reduction of NMA also explains an advanced preparation to stop. However, alternative theories can also support this reduction as they have shown that an additional working memory task can impair temporal regularity (*Maes, Wanderley & Palmer, 2015*; *Repp & Su, 2013*). For example, *Maes, Wanderley & Palmer (2015)* shows that in a SMS with a secondary working memory task, which consisted in a concurrent digit-switch counting task, the temporal regularity was highly impaired when the cognitive load was high.

Our subsequent prediction was that visual stop signals would reduce NMA to a lesser degree for vocalizations than they would for finger tapping. Our rationale for this prediction was that, in the case of finger tapping to an auditory pacing signal, the inter-modal integration between the tactile/kinaesthetic and auditory modalities allows more "room" for a perturbation to shift the synchronization timing. On the contrary, when vocalizations are synchronized to an auditory pacing signal, the intra-modal integration between the auditory modality is more likely to perceive a disruption in the accurate synchronization and therefore correct it. In other words, intra-modal integration should be more sensitive to temporal coincidence detection than inter-modal integration (*Grondin & Rousseau, 1991*; *Grondin et al., 2005*; *Sugano, Keetels & Vroomen, 2012*).

By combining the stop signal task and the SMS, *Fischer et al. (2016)* show that the NMA is significantly reduced when the possibility of a stop signal is imminent. Specifically, the inclusion of relevant stop signals induced a reduction in the NMA (for finger tapping). We predicted that such a shift in the NMA would be less tolerable in a vocalization task because synchronization in this case is underpinned by intra-modal integration.

In sum, the evidence presented so far suggests that when performing SMS tasks, the NMA is affected by differences in sensory transduction and afferent conduction between sensory modalities (e.g., the negative asynchrony of finger tapping to a tactile pacing

signal is significantly smaller compared to that for finger tapping to an auditory pacing signal). When the tap and the pacing signal occur within the same sensory modality (thus, intra-modal integration), the NMA is reduced due to equivalent sensory transduction or afferent conduction times.

To further investigate the Paillard–Fraisse modal integration hypothesis of asynchronous tapping, we investigated finger tapping and vocalizations to test two specific hypotheses. (1): that in a control condition (i.e., where synchronization was important while stopping was irrelevant), vocalizations to auditory pacing elicit shorter NMA compared to finger tapping. (2): that in an experimental condition (i.e., where both synchronization and stopping were relevant), vocalizations to auditory pacing exhibit reduced NMA, but that this reduction occur to a lesser degree than for finger tapping to auditory pacing. This lesser reduction in vocalizations occurs due to intra-modal integration that engenders more sensitive temporal coincidence detection than inter-modal integration.

## METHODS

### Participants

A total of 44 participants completed this study (32 females) aged (mean ± SD) 19.6 ± 2 years. Based on the requirement of successful stopping on about 50% of the trials for validity of the measures taken from the stop signal task (*Logan & Cowan, 1984*), data inclusion was limited to that taken from participants with a percentage of successful stopping that fell between 40% and 60% (see the description of the stop signal task below for a full explanation). A total of 14 participants did not meet this performance criterion and hence their data were excluded from the analysis. All participants were right-handed and reported no history of neurological or psychiatric conditions. Participants received course credit for their participation. The experiment was approved by Macquarie University human research ethics committee (Ref. 5201200035).

### Apparatus

The experimental task was coded in Presentation software (version 16.1, www.neurobs.com) and delivered via a Samsung monitor (SyncMaster SA950_LS27A950, 27 inches, $1,920 \times 1,080$ pixels, 120 Hz refresh rate). In audio settings, the Presentation mixer mode was DirectX. The auditory pacing signal was synthesized in Audacity (version 1.34-beta, *Audacity Team, 2018*) at 750 Hz and 75 ms in length. The auditory pacing signal was presented via Sennheiser headphones (HD 280 pro) with up to 32 dB ambient noise attenuation. Vocalizations were recorded (eight bits, two channels, 48 kHz) via an external Shure microphone (version WH20XLR) positioned two cm from each participant's mouth. The microphone signal was amplified by an ART Tube MP Pre-Amp. Vocalizations were identified via the sound response device in Presentation software, which detected a response when a sound passed a minimal threshold (set to 0.04 on a zero–one scale). This scale represents a percentage of the maximum sound recording level. Finger taps produced by the right index finger were recorded via a Cedrus RB-830 button box. Participants were seated approximately 80 cm from the monitor.

## Calibration of responses

Vocal responses were recorded and used to verify, offline, the accuracy of the response times reported by Presentation software. A Matlab script that detected a vocal response onset based on the sound envelope of the vocalization showed a high level of concordance with the software voice key. This analysis is located in the Supplementary Material.

Further to this analysis, we calibrated the response times reported by Presentation with PowerLab 8/35 (https://www.adinstruments.com/products/powerlab) that measured the veridical times of the stimulus and response relative to each other. We used this to adjust the two response modalities, subtracting 75.83 ms from the button press and 67.43 ms from the vocal responses. Please refer to "Appendix A" for a description and graphical representation of this calibration procedure.

## Stop signal task and SMS

This study combined two paradigms: the stop-signal task (SST; *Logan, 1994*; *Logan & Cowan, 1984*) and the SMS task, (*Stevens, 1886*; *Wing, 2002*). For a graphical illustration of the stop signal task and the current study's task, please see "Appendix B". This combined task consisted of two blocks: relevant and irrelevant stops. Both the relevant and irrelevant stop blocks included both go and stop trials. In the irrelevant stop block, participants were instructed to ignore the stop trials and treat them as go trials. Thus, the irrelevant stop block was very similar to the classic SMS task where participants only synchronize their responses to the pacing signal. Whereas in the relevant stop block, participants had to both synchronize to the pacing signal and to try to withhold their synchronization response whenever a stop signal appeared.

In go trials, participants were asked to synchronize to an auditory pacing signal by either finger tapping or vocalizing. Finger tapping consisted of pressing a response key in time with the auditory pacing signal, whereas vocalizations consisted of producing the vowel sound "I" as it would sound in the word "hit /hIt/" in time with the auditory pacing signal.

When detecting vocalization onsets, it is ideal to use a low threshold for the sound key in order to mitigate against delays in the onset. However, this means that there can be false onsets detected by lip noises. By using vowel sounds, participants were able to respond these sounds without having to open and close their lips. We also showed the participants how Presentation software detected the vocalizations (sound device settings) and indicated how a lip sound could be detected as a vocal response. We instructed participants to keep their lips apart during the testing and avoid lip sounds and sighs. Furthermore, because vowel sounds can be made with little jaw movement they are better for EEG/MEG studies. Because we have conducted such studies previously we use consistent stimuli here in order to be able to compare our results across experiments (*Castro-Meneses, Johnson & Sowman, 2016*; *Etchell, Sowman & Johnson, 2012*).

In stop trials, participants were asked to withhold their response when they saw a stop signal. The stop signal was visually presented as a red "X" (400 font, 12 cm height, 10.5 cm width, 8.5° visual angle and 200 ms in duration) on a black background. There was a minimum of one and a maximum of 10 go trials after any stop trial. The stop signal

was initially placed 200 ms before the pacing signal and adjusted according to a dynamic stop-signal delay (SSD) staircase (for a more extensive description of the stop signal task and a similar protocol see *Castro-Meneses, Johnson & Sowman, 2015*). Each response modality had a separate SSD staircase adjusted independently, thus there were two staircases, one for finger taps and one for vocalizations. The staircase adjustment changed the SSD after every stop trial by approximately 30 ms (to the nearest multiple of 8.3 ms as dictated by the monitor refresh rate), increasing it by 30 ms if participants successfully inhibited their previous response and decreasing it by 30 ms if previous inhibition was unsuccessful (*Logan, Schachar & Tannock, 1997*; *Osman, Kornblum & Meyer, 1986*, *1990*; *Verbruggen & Logan, 2009a*). This method aims to return a percentage of successful stopping of approximately 50%.

A block contained 80 trials, of those, 60 trials were go trials (3/4 of total trials) and 20 trials were stop trials (1/4 of total trials). The inter-onset interval between the auditory pacing signal was constant at 1,250 ms. This inter-onset interval allowed a sufficient gap between stop trials, such that the stop signals would be perceived as clearly occurring before the auditory pacing signal. This time is still within the limits of optimal synchronization performance intervals which range from about 175 ms to 1,580 ms (*Bolton, 1894*; *London, 2002*). For an illustration of the trial structure, see Fig. 1.

### Estimating asynchronies

Our first aim was to estimate the time at which the finger taps or vocalizations were produced with respect to the onset of the pacing signal. To do this, we subtracted the time of the pacing signal ($t_{ps}$) from the time of a response ($t_r$): $t_{ps}-t_r$. Positive values indicate a response occurred after the pacing signal, whereas negative values indicate a response occurred before the pacing signal.

### Estimating the perturbation of stop signals on the asynchronies: proactive inhibition

Our second aim was to measure the degree by which stop signals reduce the NMA in both finger tapping and vocalization. This corresponds to a common measure extracted from the stop signal task called proactive inhibition (see *Castro-Meneses, Johnson & Sowman, 2015* for in depth description). Proactive inhibition is defined as the amount of go RT slowing due to the introduction of stop signals. To measure this slowing requires both a control block, where stop signals are ignored or not presented, and an experimental block where stop signals are relevant; the difference in RT between these two blocks is then obtained (this is referred to block-by-block analysis, for a trial-by-trial analysis see *Verbruggen & Logan, 2009b*). To do this, we subtracted these response times from the relevant stop condition from those of the irrelevant stop condition in both finger tapping and vocalizations.

### Estimating reactive inhibition

Reactive inhibition is a measure obtained from stop signal task performance. The analysis of reactive inhibition did not have any direct impact on this study's aims but was included

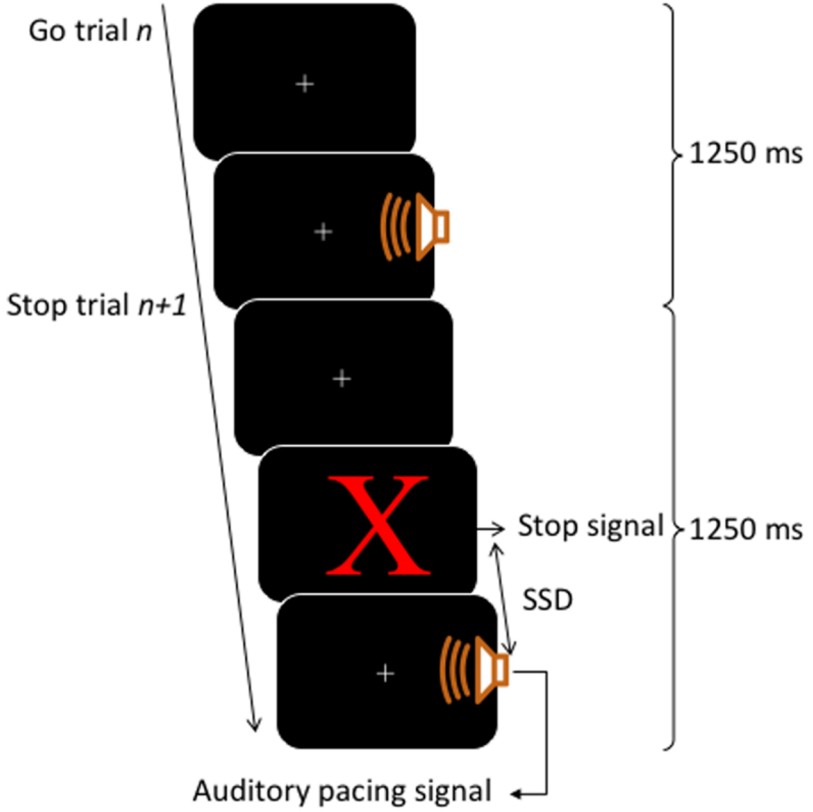

**Figure 1 Trial structure.** Each trial lasted for 1,250 ms. Go trials consisted of a visual fixation cross (continuously presented) and an auditory pacing signal: a tone (750 Hz, 75 ms in duration) emitted 400 ms into a trial. Stop trials started as per go trials but a stop signal consisting of a red X was presented on the screen before the auditory pacing signal. The stop signals were initially located in time 200 ms before the auditory pacing signal and were adjusted throughout via a staircase method in which, after a successful stop, the next stop trial was made harder by decreasing the time between the stop signals and the auditory pacing signal by 30 ms. Contrarily, if participants were unsuccessful on a stop trial, the next stop trial was made easier by increasing the time of the stop signal in relation to the pacing signal by 30 ms.

for completeness. Reactive inhibition is indexed by the SSRT and is based on the independent horse-race model (*Logan & Cowan, 1984*). We used the integration method to estimate SSRT, which is considered the most robust approach for estimating SSRT (*Verbruggen, Chambers & Logan, 2013*; *Verbruggen & Logan, 2009a*). Using this method, SSRT was calculated by subtracting the starting time of the stop process (when participants saw a stop signal) from the finishing time of the stop process. The starting time of the stop process refers to the time of the SSD. The finishing time needs to be estimated. The finishing time is usually estimated by integrating the go RT distribution. The go distribution in the current task did not contain RTs but rather go synchronized responses (go SR), thus we estimated the finishing time from the go synchronized time distribution. Synchronized responses were estimated as the time a button press or vocal response occurred within the trial. The distribution of the go SR was rank ordered from shortest to the longest then, the *n*th SR was selected, where *n* was selected by multiplying the probability of responding on stop trials (or unsuccessful stopping) by the

total number of go SR. The probability of responding was calculated as the number of unsuccessful stops divided by the total number of stop trials. SSRT was estimated by subtracting the SSD from $n$th go SR. SSRT can only be estimated in the relevant stop blocks. We first calculated it across each block and then it was averaged across blocks. A graphical representation of the variables in the classic stop signal task as well as in the current task are depicted in Fig. A6.

## Experimental design

This study involved one session divided into two phases: phase-1 and phase-2. At the end of phase-1, participants were presented with a message saying: "You are halfway through the experiment, you have four more blocks to go". Each phase consisted of four blocks (in total eight blocks) in which two blocks were relevant stop blocks (one finger tapping and the other vocalization) and two blocks were irrelevant stop blocks (one finger tapping and the other vocalization). The four-block order (i.e., a phase) was pseudo-randomized by two conditions. Firstly, a response modality could only be followed by the other response modality of the same stop block (either irrelevant or relevant stop). Secondly, a stop block could only be followed by the other stop block once the first condition was met. For example, if the first block was a relevant stop block with finger tapping then the second block had to be the same stop block (i.e., relevant stop block) with the other response modality (i.e., vocalization). Subsequently, the third and fourth blocks were irrelevant stop blocks, the third with finger taps and the fourth block with vocalizations. At the beginning of each block, instructions were presented on the screen indicating what type of stop block and response modality was to follow, see Fig. 2. The purpose of the phases was to ease the task complexity.

## Procedure

Participants were told that they had to synchronize their response in time with an auditory pacing signal and to attempt to withhold this response when a stop signal was presented. They were encouraged to relax as much as they could and respond with ease. We explained that both synchronizing and stopping were equally important and that they would fail to stop on about 50% of the stop trials because the experiment adjusted itself to give them easy and hard stop signals according to their performance. Each participant completed at least one practice task. The practice task was a much shorter version of the experimental task; it contained eight blocks of 12 trials each (four were stop trials). First, the experimenter performed the practice task as a demonstration. Then, the participant was invited to do the practice task and, on completion, their performance was assessed. If they had correctly synchronized to go trials, did not have more than five misses and got two successful and two unsuccessful stops, we administered the main task, otherwise, the practice task was repeated.

During the experimental task, the first 10 trials in each block were always go-trials (for the purpose of familiarization with the rhythm). These trials were not included in any statistical analyses.

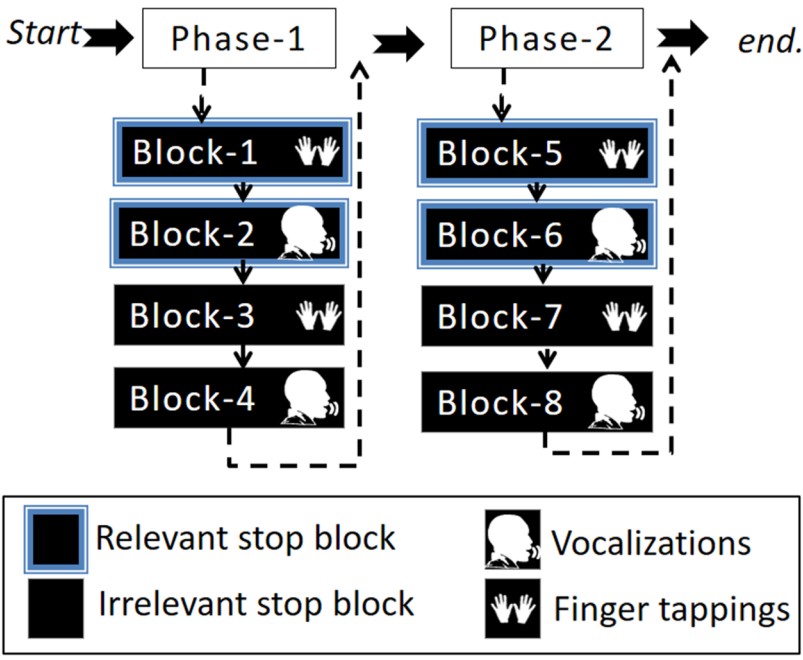

**Figure 2 Illustration of the experimental design.** The experimental task was divided into two phases that included four blocks. The four-block order (i.e., a phase) was pseudo-randomized by the condition such that two blocks were either irrelevant or relevant stop blocks for each response modality (vocalization or finger tapping).

# RESULTS

## Asynchronies

### $3 \times 2 \times 2 \times 2$ repeated-measures ANOVA

We first assessed to what degree the stop signals affected synchronization. To do this we compared the asynchronies after every stop signal and classified them as: first go after a stop signal, second go after a stop signal and all other go trials (i.e., all go trials except the first and second go). We carried out a repeated-measures $3 \times 2 \times 2 \times 2$ ANOVA on the dependent variable: NMA, with within-subject factors of three go types after stop signals (first go, second go and all other go), two stop block types (relevant stop, irrelevant stop), two response modalities (finger tapping, vocalization) and two phases (phase-1, phase-2). We were interested in the go type factor and any interaction of this factor with other factors. All results of the ANOVA are found in Appendix B and Table A1.

The results revealed three statistically significant main effects and three interactions: stop block type ($F(1, 29) = 79.31$, $p < 0.001$, $\eta_p^2 = 0.74$), response modality ($F(1, 29) = 95.80$, $p < 0.001$, $\eta_p^2 = 0.70$), phase ($F(1, 29) = 6.20$, $p = 0.019$, $\eta_p^2 = 0.18$), the two-way interaction of stop block type by response modality ($F(1, 29) = 22.60$, $p < 0.001$, $\eta_p^2 = 0.44$), the three-way interaction of go type by stop block type by response modality ($F(1.7, 48.5) = 4.59$, $p = 0.021$, $\eta_p^2 = 0.14$ Greenhouse-Geisser corrected) and the two-way interaction of go type by phase ($F(2, 58) = 5.88$, $p = 0.005$, $\eta_p^2 = 0.17$). Because we were mainly interested in the go type factor and any interaction of this factor

with other factors, we first describe the two significant interactions that involved the go type factor. A description of the main effects and interactions that were significant but did not involve the go type factor is included in Appendix C and Table A1.

Post hoc analysis following up on the two-way interaction between the go type and phase (Bonferroni corrected) revealed that the three go types were not significantly different within each phase ($ps > 0.410$) or across phases ($ps > 0.108$). The only significant difference was for the first go type across phases. Specifically, the first go (after a stop signal) in phase-1 ($M = -65.05$ ms, SE = 5.84, 95% CI [$-76.98$ to $-53.12$]) occurred 17.70 ms earlier than the first go in phase-2 ($M = -82.75$ ms, SE = 6.98, 95% CI [$-97.02$ to $-68.48$], $p < 0.001$). All other pairwise comparisons were not statistically significant; these are shown in Appendix D, Tables A2 and A3. Because there were significant differences between the first go across phases, we excluded the first go after the stop signal for the subsequent analysis in the $2 \times 2 \times 2$ ANOVA.

Post hoc analyses on the three-way interaction of go type by stop block type by response modality revealed that there were no differences across the first go, second go and all other go in finger tapping$_{relevant\ stop}$ ($ps > 0.901$), vocalization$_{relevant\ stop}$ ($ps > 0.067$), finger tapping$_{irrelevant\ stop}$ ($ps > 0.907$) and vocalization$_{irrelevant\ stop}$ ($ps > 0.999$). There were significant differences between the go types across stop block and response modality. Specifically, every go type in the relevant stop block had more positive asynchronies compared to those in the irrelevant stop block ($ps < 0.008$). Moreover, every go type of the vocalization had more positive asynchronies relative to those go types in the finger tapping ($ps < 0.003$). All pairwise comparisons of this three-way interaction are shown in Appendix E, Tables A4 and A5.

### $2 \times 2 \times 2$ repeated-measures ANOVA

After excluding the first go after the stop signals, we carried out a smaller repeated measures $2 \times 2 \times 2$ ANOVA to measure the go asynchronies. This ANOVA contained the within subject factors of two stop block types (relevant stop, irrelevant stop), two response modalities (finger tapping, vocalization) and two phases (phase-1, phase-2). The results revealed that both the factors block type and response modality were statistically significant as well as the two-way interaction between block type and response modality ($F(1, 29) = 15.03$, $p < 0.001$, $\eta_p^2 = 0.35$). Because this two-way interaction explains the two significant factors, we explain the interaction only. All other factors and interactions are described in Appendix F and Table A6. It is important to highlight that unlike the previous $3 \times 2 \times 2 \times 2$ ANOVA, the phase factor was not statistically significant ($F(1, 29) = 2.30$, $p = 0.141$, $\eta_p^2 = 0.08$). A finding that shows asynchronies were stable across the whole testing session.

Post hoc Bonferroni corrected comparisons examining the two-way interaction between stop block and response modality showed that the asynchronies in the relevant stop condition were delayed relative to those of the irrelevant stop condition ($ps < 0.001$). Specifically, asynchronies for finger tapping$_{relevant\ stop}$ ($M = -70.24$, SE = 8.62, 95% CI = [$-87.86$ to $-52.62$]) occurred 58.05 ms later compared to those

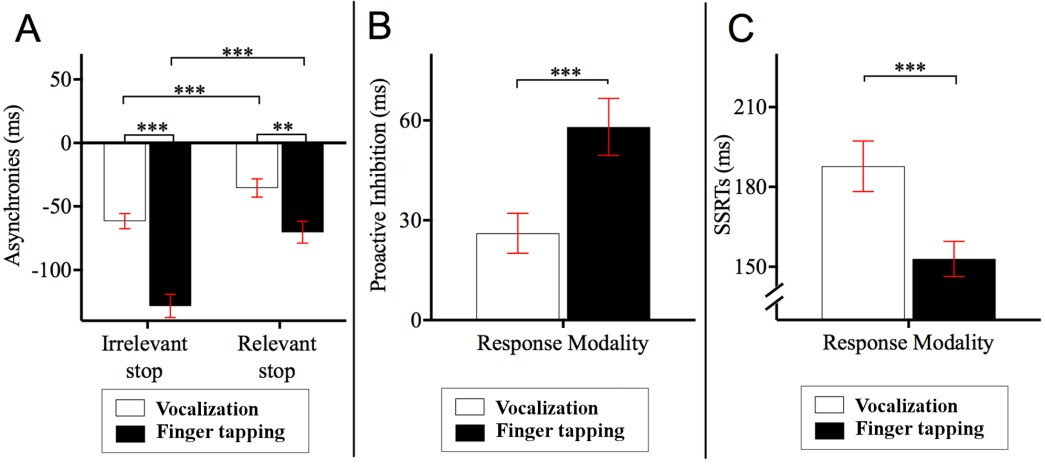

**Figure 3 Asynchronies, proactive inhibition and stop signal reaction time (SSRT).** (A) Two-way interaction between stop block type and response modality from the 2 × 2 × 2 ANOVA on the dependent variable asynchrony. Horizontal dashed line indicates onset of auditory pacing signal. Stop signals in the relevant stop condition reduced the NMA such that vocalization and finger tapping were delayed. Specifically, for finger tapping, the stop signals delayed the synchronization by 58.05 ms whereas for vocalizations, stop signals in the relevant stop condition delayed the NMA by 26.10 ms. (B) Proactive inhibition across response modalities. Proactive inhibition was estimated by subtracting the asynchrony times in the relevant stop condition from those in irrelevant stop condition. The data showed that stop signals shifted the asynchronies to a lesser degree for vocalization (26.11 ms) compared to finger tapping (58.06 ms). (C) SSRTs across response modalities. There were statistically significant differences between the SSRTs across vocalization or finger tapping. Specifically, SSRT of finger tapping was shorter (152.85 ms) compared to that for vocalizations (187.81 ms) ***$p < 0.001$, **$p < 0.01$. Error bars indicate standard error of the mean (SE).

asynchronies for finger tapping$_{\text{irrelevant stop}}$ ($M = -128.29$, SE $= 9.19$, 95% CI $= [-147.07$ to $-109.51]$). Likewise, the asynchronies of vocalization$_{\text{relevant stop}}$ ($M = -35.37$, SE $= 7.16$, 95% CI $= [-50.01$ to $-20.74]$) occurred 26.10 ms later compared to the asynchronies of vocalization$_{\text{irrelevant stop}}$ ($M = -61.47$, SE $= 5.92$, 95% CI $= [-73.57$ to $-49.37]$). This interaction also showed that the asynchronies of vocalization were significantly different to those of the finger tapping ($p$s $< 0.001$). Specifically, asynchronies for vocalization$_{\text{irrelevant stop}}$ occurred 66.82 ms later compared to those asynchronies for finger tapping$_{\text{irrelevant stop}}$. Similarly, the asynchronies for vocalization$_{\text{relevant stop}}$ occurred 34.87 ms later in comparison to the asynchronies of finger tapping$_{\text{relevant stop}}$. See Fig. 3A for an illustration of this interaction.

## Effect of stop-signals on the asynchronies: proactive inhibition

We conducted a paired sample $t$-test across response modalities for proactive inhibition. This was calculated as the time difference between relevant and irrelevant stop conditions. These time differences were obtained from the second go after stop signals and all other go after the stop signals. This analysis allowed us to measure the degree that stop signals affected synchronization (i.e., how much a participant holds back their response in anticipation of a stop signal occurring). The results showed that stop signals during vocalization delayed NMA to a lesser degree ($M = 26.11$, SE $= 5.99$, 95% CI $= [14.64–37.80]$, $p < 0.001$) compared to finger tapping ($M = 58.06$, SE $= 8.57$,

95% CI = [40.89–74.62]). In other words, stop signals did not delay vocalizations as much as they did for finger taps. This result is depicted in Fig. 3B. For an analysis of proactive inhibition taking all the go types including the first go after stop signals, second go after stop signals and all other go after the stop signals see Appendix F and Table A7. The analysis shows that the insertion or extraction of the first go after the stop signals did not alter the observation that vocalizations induced less proactive inhibition relative to finger tapping.

### Reactive inhibition—SSRTs

Reactive inhibition was indexed by the SSRT. SSRT can only be estimated for the relevant stop condition. A paired sample $t$-test showed that the SSRTs were statistically different between vocalization and finger tapping ($p < 0.001$). This difference showed that finger tapping had significantly shorter SSRT ($M = 152.95$, SE = 6.60, 95% CI = [140.46–165.60] relative to the SSRT of vocalizations ($M = 187.81$, SE = 9.52, 95% CI = [170.74–205.56]). An illustration of this analysis is depicted in Fig. 3C.

### Descriptive statistics

A table of the descriptive statistics of go asynchronies, stop signal delay, SSRT and go misses can be found in Appendix G and Table A8.

### Checking assumption of the horse race model

For completeness of this study, we checked the assumption of the horse race model that says that in a failed stop trial, failed stop RT are shorter relative to the go RT. A finding that indicates the go process won the race against the stop process (*Logan & Cowan, 1984*). To see this analysis, please go to Appendix H.

## DISCUSSION

The current study investigated a classic finding that occurs in finger tapping to an auditory pacing signal: that taps occur prior to the auditory pacing signal (*Dunlap, 1910*; *Johnson, 1899*; *Wallin, 1904*). An explanation for this phenomenon comes from the Paillard–Fraisse hypothesis (*Aschersleben & Prinz, 1995*) and the sensory modal integration theory (*Müller et al., 2008*) that both conclude that finger tapping to an auditory pacing signal requires two sensory modalities: the tactile/kinaesthetic and auditory. Because the sensory transduction and afferent conduction times vary between these two sensory modalities, there is an inherent offset in the arrival times of these signals in the brain that causes the NMA. Here, we addressed the question: if the afferent signal from the pacing signal and the reafferent signal caused by responding are elicited within the same sensory modality, would the NMA be reduced? This study compared both vocalization and finger tapping to an auditory pacing signal. Because sensory integration would occur in only the auditory modality, it was hypothesized that the NMA would be less for vocalization relative to finger tapping. Secondly, it was hypothesized that stop signals would reduce asynchronies, but to a lesser degree in the vocalization compared to the finger tapping. Our data support both of our hypotheses, these are discussed next.

The first hypothesis stated that in a control condition where synchronization was important while stopping was irrelevant, the negative asynchrony is smaller for vocalization to an auditory pacing signal compared to the finger tapping to an auditory pacing signal. To test this hypothesis, the control condition had stop signals that were irrelevant. This control condition was called the irrelevant stop condition. We found that while the NMA of the finger tapping occurred at −128.29 ms, those of the vocalizations occurred at −61.47 ms. In other words, finger tapping had 66.82 ms more negative asynchrony compared to vocalizations.

These results support the behavioral distinctions that result from the differences between intra- and inter-modal integration described in *Müller et al. (2008)*. The larger NMA in finger tapping can be explained by the requirement to compare event timing across two sensory modalities. The differences in sensory transduction and afferent conduction times across the sensory modalities for the inter-modal integration produce larger NMA. On the other hand, the shorter NMA for vocalizations is explained by the requirement to compare sensory information within one sensory modality. *Müller et al. (2008)* found that both finger and toe tapping to a tactile pacing signal (thus an intra-modal integration) had NMA of less than 8 ms but these differences were not significantly different from the onset of the tactile pacing signal. Whereas both finger and toe tapping to an auditory pacing signal (thus, an inter-modal integration) produced negative asynchronies of about −40 ms that were significantly negative relative to the onset of the auditory pacing signal.

Our second hypothesis, following on from *Fischer et al. (2016)* stated that the stop signals in the relevant stop blocks reduce the NMA, but this reduction occurs to a lesser degree for vocalization relative to finger tapping. We found that asynchronies were reduced in the relevant stop blocks compared to the irrelevant stop block in both the vocalization as well as in the finger tapping. Specifically, from the irrelevant to the relevant stop, asynchronies of the finger tapping decreased from −128.29 ms to −70.24 ms whereas asynchronies of the vocalization decreased from −61.47 ms to −35.37 ms. These results show that stop signals induced proactive inhibition, thereby delaying the onset of the synchronization response and reducing the NMA.

The next part of this hypothesis was to investigate whether stop signals reduced the NMA to a lesser degree for the intra-modal integration case (i.e., vocalizations to auditory pacing signal) relative to the inter-modal integration case (i.e., finger tapping to auditory pacing signal). The analysis of proactive inhibition showed that for vocalizations, responses were slowed (proactive inhibition) by 26.10 ms (vocalizations$_{relevant\ stop}$ − vocalizations$_{irrelevant\ stop}$) compared to 58.05 ms for finger tapping (finger tapping$_{relevant\ stop}$ − finger tapping$_{irrelevant\ stop}$). This finding suggests that vocalization was less perturbed by the stop signals compared to finger tapping. This supports the contention that sensory integration that occurs between modalities (i.e., inter-modal integration) contains a degree of "play", such that a broader range of tap timings might be tolerated compared to sensory integration within the same sensory modality (i.e., intra-modal integration. *Grondin & Rousseau, 1991*; *Grondin et al., 2005*).

Given that we found that the NMA for the intra-modal integration was less affected by stop signals compared to inter-modal integration, further studies should assess whether this stability can also be extended to performance variability. In other words, whether intra-modal integration has less tap variability than inter-modal integration. Research suggests that SMS is controlled independently by two error correction processes (phase correction and period correction). Phase correction is considered an automatic process that does not interfere with the tempo of tapping whereas period correction is usually intentional and changes the tempo (see *Repp, 2005*; *Repp & Su, 2013* for reviews). This study also opens up research on delayed feedback in which temporal recalibration is observed when taps become more negative (*Cunningham, Billock & Tsou, 2001*; *Heron, Hanson & Whitaker, 2009*; *Stetson et al., 2006*; *Sugano, Keetels & Vroomen, 2012*).

An interesting aspect to highlight here is that stop signals in the relevant stop blocks made finger tapping and vocalizations more synchronous. This finding shows that stop signals, by inducing proactive inhibition, decreased the NMA of finger tapping and vocalizations—this increment in proactive inhibition improved synchronization. Evidence showing that proactive inhibition plays a crucial role in more accurate synchronizations comes from studies that use explicit performance feedback during SMS. In these experiments, when participants are informed about the size and the direction of the asynchrony (knowledge of results), they are able to tap in exact physical synchrony. However, participants in such experiments report that they have to delay their response in order to perform with greater accuracy (see *Aschersleben, 2002*, pp. 67–68). Unlike musically untrained people, trained musicians show significantly less NMA (~−14 ms) suggesting that musical training may require learning to delay the response so it can occur in time (*Aschersleben, 2002*; *Repp, 2004*). The link that connects such observations and the current study is that the control of response releases in SMS tasks is likely due to the application of inhibitory control.

The idea that inhibitory control and rhythmic timing are linked is not a new one. A number of studies note the association between rhythmic ability and executive control dysfunctions, or stages of executive control development (for a review see *Repp & Su, 2013*; *Rubia et al., 1999*). Furthermore, there is a significant overlap in the neural substrates that support inhibitory control and the production of rhythmic tapping tasks (*Aron, Robbins & Poldrack, 2004*; *Aron, Robbins & Poldrack, 2014*; *Buhusi & Meck, 2005*; *Doumas, Praamstra & Wing, 2005*; *Gross et al., 2002*; *Jäncke et al., 2000*; *Koch, Oliveri & Caltagirone, 2009*; *Lewis & Miall, 2003*; *Middleton & Strick, 2000*; *O'Boyle, 1999*; *Wiener, Turkeltaub & Coslett, 2010*; *Witt & Stevens, 2013*). More directly addressing this observation, *Witt & Stevens (2013)* show that top-down influences onto the motor cortex from dorsal and ventral prefrontal cortices—particularly in the right hemisphere (*Wiener, Turkeltaub & Coslett, 2010*)—are associated with the performance of rhythmic tapping. These same areas of the right hemisphere are also strongly associated with inhibitory control (see *Aron, Robbins & Poldrack, 2004*; *Aron, Robbins & Poldrack, 2014*). Moreover, it has previously been demonstrated that longer intervals SMS is related to higher activation in the right dorsolateral pre-frontal cortex (*Koch, Oliveri & Caltagirone, 2009*). A transcranial magnetic study found that cortical inhibition to the motor and premotor

cortex reduced the NMA (*Doumas, Praamstra & Wing, 2005*). Such studies support our contention that inhibitory control is positively related to accuracy of SMS.

An incidental finding worth highlighting in this study is that SSRTs were significantly longer for vocalizations relative to finger tapping. This finding is in line with previous studies that have found vocal responses in the stop signal task exhibit longer SSRTs compared to finger responses (*Castro-Meneses, Johnson & Sowman, 2015*), or that there is a trend towards this being the case (*Castro-Meneses, Johnson & Sowman, 2016*; *Wessel & Aron, 2014*). *Castro-Meneses, Johnson & Sowman (2016)* presented evidence to suggest that the vocalization system may have weaker reactive inhibition (index by the SSRTs) compared to that of the finger system, which they attributed to inhibitory control differences at the motoneuronal level (*Sowman et al., 2008*).

One of the limitations of this study is that we assumed that the vocalizations to the auditory pacing signal occurred within the same auditory modality (thus, a case of intra-modal integration). However, the vocalization response can also have tactile/ kinaesthetic sensory information when the muscles of the mouth move. We acknowledge this and believe that this could actually explain why the mean negative asynchrony between the vocalization and the auditory pacing signal are not smaller as in *Müller et al. (2008)*. In our present study, vocalizations had a NMA of 61.47 ms in the irrelevant stop condition, whereas in the intra-modal integration task of *Müller et al. (2008)* between finger and toe tapping to a tactile pacing signal had a mean negative asynchrony of approximately 8 ms. It could be that the asynchronies of the vocalizations are the average of two sensory modalities: the auditory modality and the tactile/kinaesthetic modality. This idea then would support the sensory accumulator hypothesis (*Aschersleben, 2002*) that suggests the NMA is a result of a central accumulator of different sensory channels.

In support to the sensory accumulator hypothesis, *Aschersleben & Prinz (1995)* showed that finger and foot tapping to an auditory pacing signal with auditory feedback (i.e., auditory feedback was presented in time with a tap) reduced the negative asynchrony of both finger and foot tapping significantly more compared to the same tasks without auditory feedback. The authors explained this reduction of NMA via the joint-event hypothesis, in which they said that, because the taps with auditory feedback carried two channels of sensory information: tactile/kinaesthetic and auditory, the brain averages the conduction and sensory transduction times of these two sensory modalities and that is why the NMA reduced but did not disappear completely.

In the case of our finger tapping to auditory pacing signal, the action of pressing the key button could have also elicited a sound, which could have provided auditory feedback within the same sensory modality as the auditory pacing signal. However, this auditory feedback was reduced because we asked participants to press the key lightly and they also wore ambient noise attenuating headphones (see Methods section).

We also want to emphasize that while we have taken significant steps to ensure adequate accuracy of the reported timing data across modalities (see Appendix A) a level of timing uncertainty remains inherent in our data. Future studies along these lines would be best served by recording the outputs (auditory beats) in parallel with the inputs (microphone and button press) using synchronized DAQ channels in a dedicated

recording device. In this way absolute veridical timing relationships can be obtained between inputs and outputs (*Schultz & Van Vugt, 2016*). Where possible button presses should also be recorded with integrated force transducers in series so that button press timing is not a function of computer port polling. Such methods circumvent the need for complex calibration procedures and ensure the optimal timing accuracy is obtained. Timing accuracy should be considered a limitation of our findings that vocalization exhibits less NMA than finger tapping. However, the interaction between modality (vocal vs. manual) and stop condition (relevant vs. irrelevant) is unaffected by this limitation.

In conclusion, the current study investigated negative asynchronies of finger tapping and vocalizations to an auditory pacing signal in a modified stop signal task context. When stopping was not required, the data show that vocalization were produced with smaller NMA compared to finger tapping. When stopping was required, proactive inhibition induced a delay in the responses, thereby reducing the NMA. Importantly, this decrement was smaller for vocalizations than for finger tapping. Our data support the Paillard–Fraisse hypothesis, which predicts that negative asynchrony should be smaller for vocalization because this is achieved via an intra-modal integration. We extend this hypothesis by contending that intra-modal integration should be more sensitive to synchronization discrepancies and therefore would be more resistant to perturbation. Our data show that stop signals did not perturb the synchronization response of vocalization as much they did for finger tapping.

## APPENDIX A. CALIBRATION METHOD

As stated in the Methods section, we use the Presentation mixer mode of DirectX that has a playback delay of about 50 ms. Figure A1 shows a graphical representation of the experimental circuits.

We used PowerLab 8/35 to measure the veridical times of inputs and outputs. (https://www.adinstruments.com/products/powerlab). We conducted tests with a sampling rate of 1 Khz which would mean a possible error in accuracy of ~1 ms. Testing was conducted under the same lab conditions as the report.

### Step 1: Time differences between output and input A (vocalizations) recorded in PowerLab

In the first step, we measured timing in the electronic part of the loop. By triggering the scope function in PowerLab we recorded simultaneously both output and input A (as depicted in Fig. A2). The microphone of input A recorded the same output (i.e., the beat). We found a timing difference on average of less than 1 ms (below the sampling accuracy). *We also conducted the same procedure with an oscilloscope—the results were the same, and for analysis purposes the output from the PowerLab is easier to deal with*.

### Step 2: Time differences between output and input A (vocalizations) recorded in Presentation software

As depicted in Fig. A3, in Presentation software we compared the time differences of the output and input A, when input A was recording the same output (i.e., the beat).
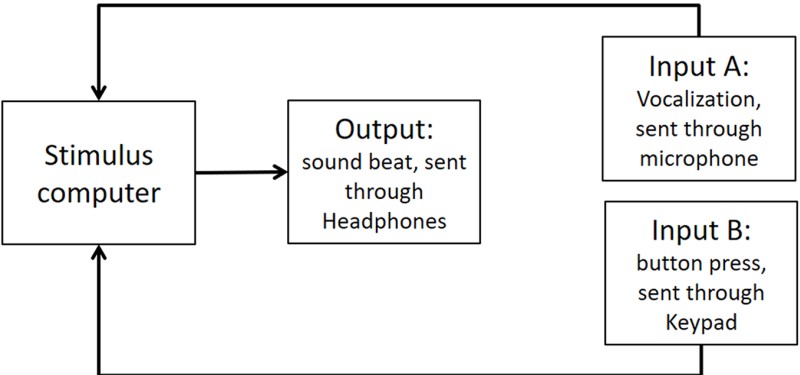

**Figure A1 Experimental circuit representing output and inputs relative to stimulus computer.** Output refers to the beat sound. It was elicited through headphones via the stimulus computer's soundcard. The inputs represent response modalities. Input A refers to vocalizations. These were picked up by a microphone and sent back to the stimulus computer to be recorded. The Input B indicates finger taps. These responses were picked up by a keypad and sent back to the stimulus computer.

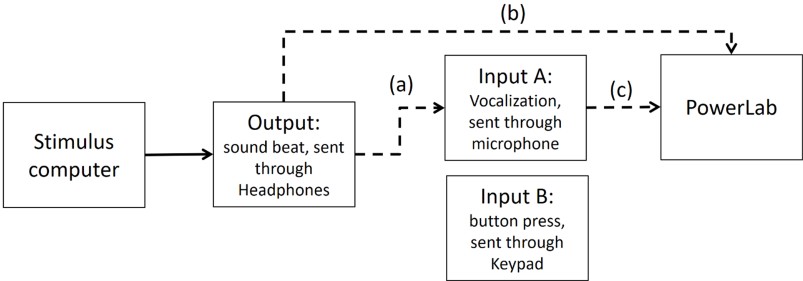

**Figure A2 Output and Input A recorded in PowerLab.** The dashed line (A) shows that Input A is recording the same output sound. The dashed lines (B and C) show that the output and Input A respectively are recorded in PowerLab simultaneously. PowerLab recorded a timing difference of less than 1 ms between (A) and (B).

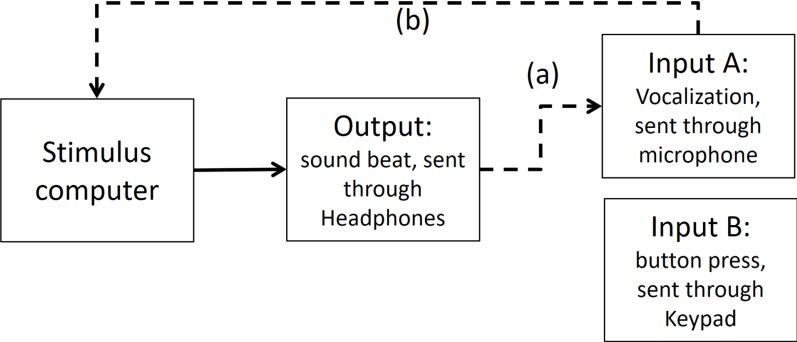

**Figure A3 Output and Input A recorded in Presentation Software.** The dashed line (A) shows that Input A is recording the output sound. The dashed line (B) shows that Input A is recorded in stimulus computer.

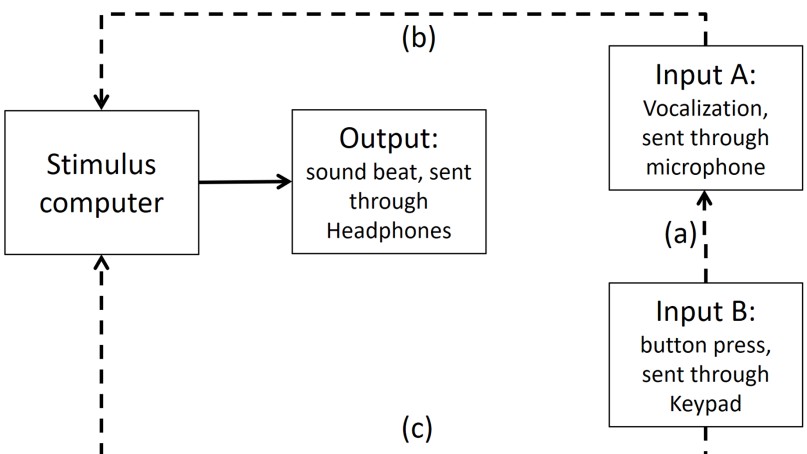

**Figure A4 Input A recording Input B in Presentation Software.** The dashed line (A) shows that Input A is recording the sound emitted from Input B. The dashed lines (B and C) show that both inputs are recorded in stimulus computer.

We found that from the time the stimulus computer—specifically the experimental software elicited an output, 67.43 ms later (range between 72.5 and 62.2 ms) input A was recorded in the stimulus computer.

We know from Step 1 that the time delay between the output and input A in PowerLab is less than 1 ms, so this delay can be attributed to a difference in experimental stimulus time vs. the actual output time from the stimulus computer's soundcard.

This meant that to adjust the recorded time of the vocalizations in Presentation software to a more veridical time, we needed to subtract 67.43 ms of the vocalizations.

## Step 3: Time differences between input A (vocalizations) and input B (finger taps) recorded in Presentation software

To test if any significant timing difference existed between input B (button press) and input A (vocalization), the sound of the button press was enhanced by striking the button hard enough to make a recordable response. Then, input B recorded the sound elicited of a button press (see Fig. A4 for an illustration of this).

Times recorded for the input B (button press) and the input A (sound) were minimally different (average of 8.4 ms, range between 4.99 and 10.36 ms). The recorded time of the button press (input B) was registered in average 8.4 ms later than the microphone (input A). This delay in input B is very close to the manufacturer's latency claim of ~4 ms for the Cedrus keypad device. The extra time is likely due to the mechanical coupling time, which we assume the manufacturer does not count.

This means that to calibrate input A and B, we needed to subtract 8.4 ms of the button press (input B) to make it comparable to input A (vocalizations). Moreover, in Step 2 we already found that input A (vocalizations) had a delayed time of 67.43 ms. This means that further 67.43 ms would be need to be subtracted from input B (button press).

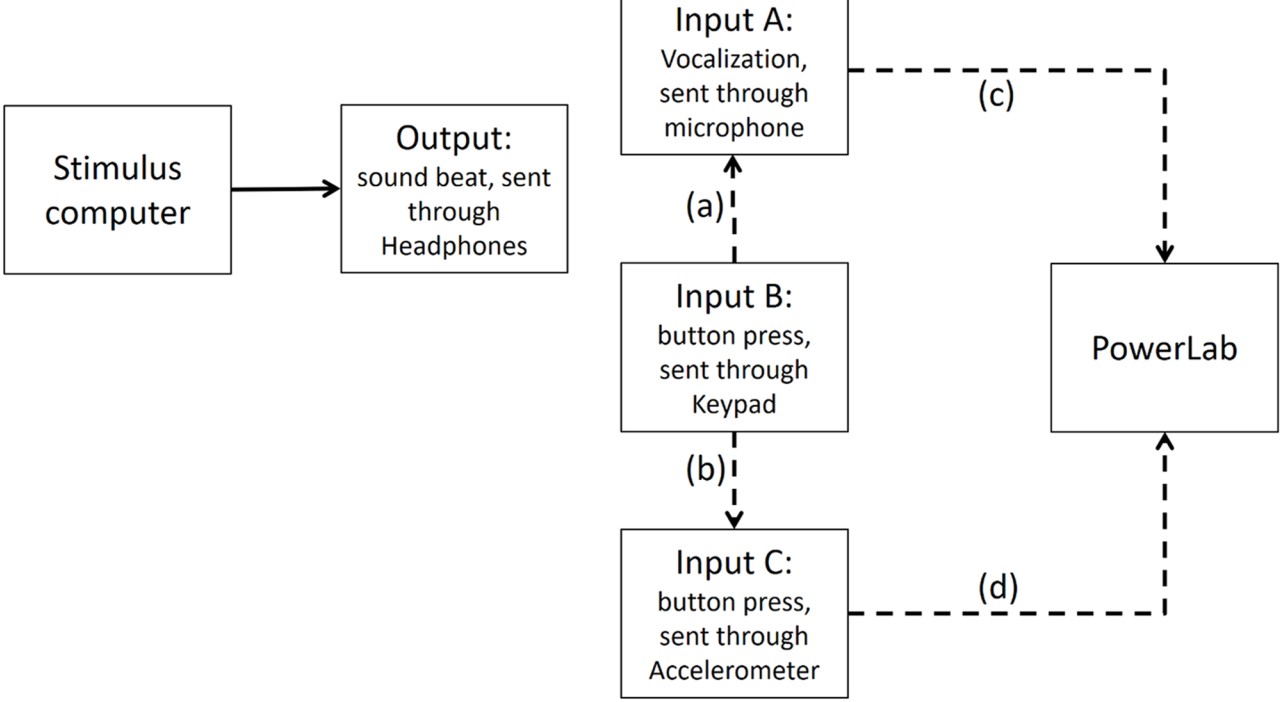

**Figure A5** **Input B is recorded via both input A and Input C (i.e., accelerometer) in PowerLab.** The dashed line (A) shows that Input A (i.e., the microphone) is recording the sound emitted from Input B (i.e., the button press). The dashed line (B) indicates that Input C, which is the accelerometer, records activity of Input B. The dashed lines (C and D) show Input A and Input C are recorded in PowerLab simultaneously. The results show that a button press recorded through an accelerometer and the sound emitted from it had less than 1 ms timing difference in PowerLab.

### Step 4: Time differences between input A (vocalizations) and accelerometer mounted on input B (button press) recorded in PowerLab

We further calibrated the method of recording Input B (button press) via the input A (microphone picking button press' sound) and via an accelerometer mounted on the button (input C as shown on Fig. A5). We used the accelerometer input to trigger the scope and record the sound of input A on the second channel. The time difference between the accelerometer and Input A was in the order of 1 ms, indicating that the microphone input was a very good proxy for the mechanical contact time.

### Conclusion

To calibrate the two response modalities, we subtracted 8.4 ms of the button presses to make them comparable to vocalizations. Moreover, to adjust for the time delay between when a response was given and the time that response was registered in presentation, we subtracted 67.43 ms from both vocalizations and finger taps.

## APPENDIX B. COMPARISON OF THE HORSE RACE MODEL WITH THE CURRENT STUDY'S TASK

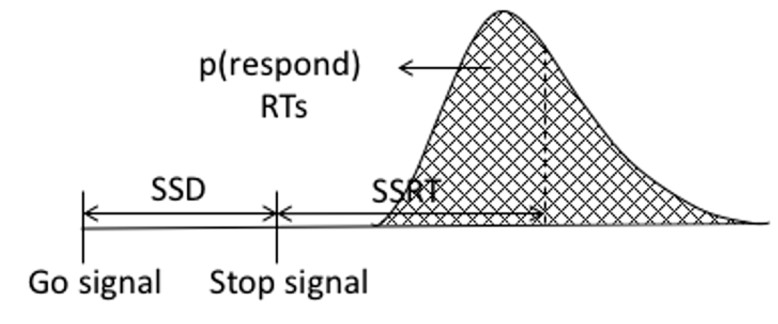

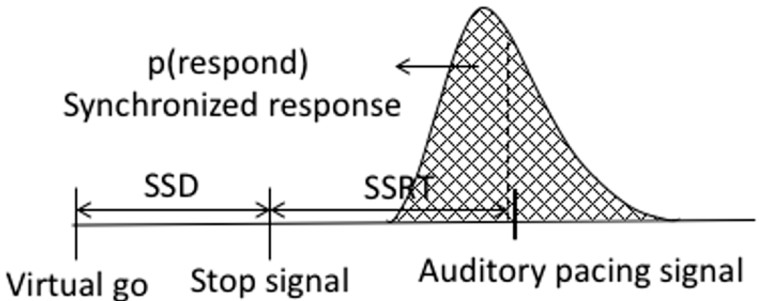

**Figure A6 Graphical representation of the independent horse-race model and the current study's task.** (A) Graphical representation of the traditional stop-signal task (*Logan & Cowan, 1984*). The graph shows that a go signal triggers a go-response while a stop signal triggers withholding of the go-response. The responses are a reaction time, which are considered the end of the go-response. The area under the curve to the left of the dashed line represents probability of responding in stop trials [$p$(respond)] or unsuccessful stopping. (B) Variables of the stop signal task combined with a synchronization task. In this task, there is no go signal, but a virtual go signal is estimated that must trigger the initiation of the synchronized response. It is inserted post hoc to allow calculation of the SSRT. A stop signal triggers withholding of the go-response. Synchronized responses consisted of finger taps or a vocalization. These are considered the end of the go.

## APPENDIX C. SIGNIFICANT FACTORS AND INTERACTIONS OF THE 3 × 2 × 2 × 2 ANOVA

Following up from the results section, here we describe the factors and interactions that were significant but did not involve the go type factor for the 3 × 2 × 2 × 2 ANOVA. There were three significant factors and one interaction: stop block type, response modality, phase and the interaction between stop block type and response modality. Because the interaction describes two factors (stop block and response modality), we describe the phase factor and the interaction only.

**Table A1  3 × 2 × 2 × 2 ANOVA of the go asynchronies.**

| Factor and interaction | F value (degrees of freedom) | p-value | Partial η² |
|---|---|---|---|
| Three go types | $F(1.6, 43.7) = 0.16$◆ | 0.793 | 0.01 |
| Two stop block types | $F(1, 29) = 79.31$ | **<0.001** | 0.74 |
| Two response-modalities | $F(1, 29) = 95.80$ | **<0.001** | 0.77 |
| Two phases | $F(1, 29) = 6.20$ | **0.019** | 0.18 |
| Go type × stop block type | $F(1.3, 36.1) = 0.03$◆ | 0.915 | 0.01 |
| Go type × response-modality | $F(1.5, 43.5) = 0.90$◆ | 0.390 | 0.01 |
| Stop block type × response-modality | $F(1, 29) = 22.60$ | **<0.001** | 0.44 |
| Go type × stop block type × response modality | $F(1.7, 48.5) = 4.59$◆ | **0.021** | 0.14 |
| Go type × phases | $F(2, 58) = 5.88$ | **0.005** | 0.17 |
| Stop block type × phases | $F(1, 29) = 0.05$ | 0.830 | 0.01 |
| Go type × stop block type × phases | $F(2, 58) = 1.45$ | 0.245 | 0.05 |
| Response modality × phases | $F(1, 29) = 0.74$ | 0.399 | 0.03 |
| Go type × response modality × phases | $F(2, 58) = 2.98$ | 0.059 | 0.10 |
| Stop block type × response modality × phases | $F(1, 29) = 0.24$ | 0.629 | 0.01 |
| Go type × stop block type × response modality × phases | $F(2, 58) = 0.47$ | 0.631 | 0.02 |

**Note:**
◆ Greenhouse-Geisser corrected.
Significant p-values are bold.

The phase factor showed that the asynchronies in phase-1 ($M = -68.64$, SE = 5.69, 95% CI = [−80.27 to −57.02]) occurred closer to the pacing signal compared to those asynchronies of phase-2 ($M = -79.53$, SE = 6.95, 95% CI = [−93.75 to −65.32]). This phase factor finding is also explained by the interaction between go type and phases described in the manuscript, in which the longer asynchronies in phase-1 were driven by the first go-trial after stop signals.

The interaction between stop block type and response modality showed that asynchronies from the relevant stop block occurred later compared to those of the irrelevant stop block across each response modality as follows: asynchronies of finger tapping$_{relevant stop}$ ($M = -69.23$, SE = 6.92, 95% CI = [−83.36 to −55.09]) occurred later than those from finger tapping$_{irrelevant stop}$ ($M = -129.32$, SE = 8.95, 95% CI = [−147.62 to −111.02], $p < 0.001$). Asynchronies of vocalization$_{relevant stop}$ ($M = -36.51$, SE = 6.89, 95% CI = [−50.58 to −22.43]) occurred later than those asynchronies of vocalization$_{irrelevant stop}$ ($M = -61.30$, SE = 5.47, 95% CI = [−72.49 to −50.11], $p < 0.001$). Across response modalities, this interaction showed that asynchronies of finger tapping were significantly more negative compared to those asynchronies of vocalizations in the relevant stop blocks ($p < 0.001$) as well as in the irrelevant stop blocks ($p < 0.001$). This same finding was found in the smaller 2 × 2 × ANOVA.

To sum up, we carried out this lengthy and complex 3 × 2 × 2 × 2 ANOVA to verify whether stop signals appearance interfered with the synchronization. We did not want to present data where participants lost the pacing rhythm because then, this would not represent a SMS effect. The data from the interaction between go type and phase revealed that the first go in phase-1 was significantly different from the second go and all other go across phases 1 and 2. This was the reason for which we excluded the first go to carry out a simpler 2 × 2 × 2 ANOVA.

# APPENDIX D. DESCRIPTIVE STATISTICS AND PAIRWISE COMPARISONS OF THE TWO-WAY INTERACTION BETWEEN GO TYPE AND PHASE OF THE 3 × 2 × 2 × 2 ANOVA

**Table A2 Descriptive statistics for the two-way interaction between go type and phase of the 3 × 2 × 2 × 2 repeated measures ANOVA on the asynchronies.**

| Go type | Phase 1 | | | Phase 2 | | |
|---|---|---|---|---|---|---|
| | Mean | SE | 95% CI | Mean | SE | 95% CI |
| First go | −65.05 | 5.84 | −76.98 to −53.12 | −82.75 | 6.98 | −97.02 to −68.48 |
| Second go | −70.89 | 6.22 | −83.60 to −58.19 | −79.55 | 7.45 | −94.78 to −64.33 |
| All other go | −69.99 | 6.30 | −82.87 to −57.11 | −76.29 | 7.72 | −92.07 to −60.52 |

**Note:**
SE, standard error of the mean; 95% CI, 95% confidence intervals.

**Table A3 Pairwise comparisons for the two-way interaction between go type and phase of the 3 × 2 × 2 × 2 repeated measures ANOVA on the asynchronies.**

| Pairwise comparison | p-value |
|---|---|
| First go_phase_1 vs. Second go_phase_1 | 0.409 |
| First go_phase_1 vs. All other go_phase_1 | 0.930 |
| Second go_phase_1 vs. All other go_phase_1 | 0.999 |
| First go_phase_2 vs. Second go_phase_2 | 0.999 |
| First go_B2 vs. All other go_phase_2 | 0.546 |
| Second go_B2 vs. All other go_phase_2 | 0.946 |
| First go_phase_1 vs. First go_phase_2 | **<0.001** |
| Second go_phase_1 vs. Second go_phase_2 | 0.109 |
| All other go_phase_1 vs. All other go_phase_2 | 0.213 |

**Note:**
Significant p-value is bold.

# APPENDIX E. DESCRIPTIVE STATISTICS AND PAIRWISE COMPARISONS OF THE THREE-WAY INTERACTION BETWEEN GO TYPE, STOP BLOCK AND RESPONSE MODALITY OF THE 3 × 2 × 2 × 2 ANOVA

**Table A4 Descriptive statistics of the three-way interaction between go type, stop block and response modality of the 3 × 2 × 2 × 2 repeated measures ANOVA on the asynchronies.**

| Go type | Relevant stop | | | Irrelevant stop | | |
|---|---|---|---|---|---|---|
| | Mean | SE | 95% CI | Mean | SE | 95% CI |
| First go_finger tapping | −69.07 | 7.19 | −83.76 to −54.38 | −129.94 | 9.05 | −148.45 to −111.43 |
| Second go_finger tapping | −65.68 | 8.46 | −82.98 to −48.38 | −131.61 | 9.96 | −152.02 to −111.29 |
| All other go_finger tapping | −72.94 | 9.67 | −92.71 to −53.16 | −126.37 | 9.26 | −145.3 to −107.43 |
| First go_vocalization | −36.09 | 8.10 | −52.63 to −19.54 | −60.50 | 5.19 | −71.11 to −49.90 |
| Second go_vocalization | −41.66 | 7.40 | −56.78 to −26.55 | −61.89 | 5.22 | −72.56 to −51.23 |
| All other go_vocalization | −31.77 | 7.40 | −46.89 to 16.65 | −61.50 | 6.38 | −74.53 to −48.46 |

**Note:**
SE, standard error of the mean; 95% CI, 95% confidence intervals.

**Table A5 Pairwise comparisons for the three-way interaction between go type, stop block and response modality of the 3 × 2 × 2 × 2 repeated measures ANOVA on the asynchronies.**

| Pairwise comparison | p-value |
|---|---|
| Differences across go types, and within response modality and stop block | |
| First go$_{\text{finger tapping\_relevant stop}}$ vs. Second go$_{\text{finger tapping\_relevant stop}}$ | 0.999 |
| First go$_{\text{finger tapping\_relevant stop}}$ vs. All other go$_{\text{finger tapping\_relevant stop}}$ | 0.999 |
| Second go$_{\text{finger tapping\_relevant stop}}$ vs. All other go$_{\text{finger tapping\_relevant stop}}$ | 0.902 |
| First go$_{\text{vocalization\_relevant stop}}$ vs. Second go$_{\text{vocalization\_relevant stop}}$ | 0.856 |
| First go$_{\text{vocalization\_relevant stop}}$ vs. All other go$_{\text{vocalization\_relevant stop}}$ | 0.999 |
| Second go$_{\text{vocalization\_relevant stop}}$ vs. All other go$_{\text{vocalization\_relevant stop}}$ | 0.067 |
| First go$_{\text{finger tapping\_irrelevant stop}}$ vs. Second go$_{\text{finger tapping\_irrelevant stop}}$ | 0.999 |
| First go$_{\text{finger tapping\_irrelevant stop}}$ vs. All other go$_{\text{finger tapping\_irrelevant stop}}$ | 0.999 |
| Second go$_{\text{finger tapping\_irrelevant stop}}$ vs. All other go$_{\text{finger tapping\_irrelevant stop}}$ | 0.908 |
| First go$_{\text{vocalization\_irrelevant stop}}$ vs. Second go$_{\text{vocalization\_irrelevant stop}}$ | 0.999 |
| First go$_{\text{vocalization\_irrelevant stop}}$ vs. All other go$_{\text{vocalization\_irrelevant stop}}$ | 0.999 |
| Second go$_{\text{vocalization\_irrelevant stop}}$ vs. All other go$_{\text{vocalization\_irrelevant stop}}$ | 0.999 |
| Differences across stop block, and within response modality and go type | |
| First go$_{\text{finger tapping\_relevant stop}}$ vs. First go$_{\text{finger tapping\_irrelevant stop}}$ | **<0.001** |
| First go$_{\text{vocalization\_relevant stop}}$ vs. First go$_{\text{vocalization\_irrelevant stop}}$ | **0.002** |
| Second go$_{\text{finger tapping\_relevant stop}}$ vs. Second go$_{\text{finger tapping\_irrelevant stop}}$ | **<0.001** |
| Second go$_{\text{vocalization\_relevant stop}}$ vs. Second go$_{\text{vocalization\_irrelevant stop}}$ | **0.007** |
| All other go$_{\text{finger tapping\_relevant stop}}$ vs. All other go$_{\text{finger tapping\_irrelevant stop}}$ | **<0.001** |
| All other go$_{\text{vocalization\_relevant stop}}$ vs. All other go$_{\text{vocalization\_irrelevant stop}}$ | **<0.001** |
| Differences across stop blocks, and within response modality and go type | |
| First go$_{\text{finger tapping\_relevant stop}}$ vs. First go$_{\text{vocalization\_relevant stop}}$ | **<0.001** |
| First go$_{\text{finger tapping\_irrelevant stop}}$ vs. First go$_{\text{vocalization\_irrelevant stop}}$ | **<0.001** |
| Second go$_{\text{finger tapping\_relevant stop}}$ vs. Second go$_{\text{vocalization\_relevant stop}}$ | **0.002** |
| Second go$_{\text{finger tapping\_irrelevant stop}}$ vs. Second go$_{\text{vocalization\_irrelevant stop}}$ | **<0.001** |
| All other go$_{\text{finger tapping\_relevant stop}}$ vs. All other go$_{\text{vocalization\_relevant stop}}$ | **<0.001** |
| All other go$_{\text{finger tapping\_irrelevant stop}}$ vs. All other go$_{\text{vocalization\_irrelevant stop}}$ | **<0.001** |

**Note:**
Significant p-values are bold.

# APPENDIX F. FACTORS AND INTERACTIONS OF THE 2 × 2 × 2 ANOVA

**Table A6 2 × 2 × 2 ANOVA of the go asynchronies.**

| Factor and interaction | F value (degrees of freedom) | p-value | Partial $\eta^2$ |
|---|---|---|---|
| Two stop block types | $F(1, 29) = 46.97$ | **<0.001** | 0.62 |
| Two response-modalities | $F(1, 29) = 96.98$ | **<0.001** | 0.77 |
| Two phases | $F(1, 29) = 2.30$ | 0.141 | 0.08 |
| Stop block type × response-modality | $F(1, 29) = 15.03$ | **<0.001** | 0.35 |
| Stop block type × phases | $F(1, 29) = 0.14$ | 0.717 | 0.01 |
| Response modality × phases | $F(1, 29) = 0.15$ | 0.707 | 0.01 |
| Stop block type × response modality × phases | $F(1, 29) = 0.69$ | 0.414 | 0.03 |

**Note:**
Significant p-values are bold.

## APPENDIX G. RESULTS OF PAIRED-SAMPLE *T*-TEST OF PROACTIVE INHIBITION WITH ALL GO SYNCHRONIZATION DISTRIBUTION

This analysis was done on all go asynchronies; this means first go after stop signals, second go after stop signals and all other go after stop signals. The analysis presented in the main text corresponded to only go asynchronies of second go after stop signals and all other go after stop signals. We decided to include this analysis to show that the exclusion of the first go after stop signals did not change the results.

**Table A7** **Results of paired-sample *t*-test of proactive inhibition.**

|  | $M \pm$ SE [95% CI] |
|---|---|
| Vocalization | 25.51 ± 5.78 [14.70–36.52] |
| Finger tapping | 59.18 ± 6.63 [46.32–72.36] |
| $t(29) = 4.50$, $p < 0.001$ | |

Note:
   M, mean in milliseconds; SE, Standard error; 95% CI, 95% confidence intervals.

## APPENDIX H. DESCRIPTIVE STATISTICS

**Table A8** **Descriptive statistics of go asynchronies, stop-signal delay (SSD), stop-signal reaction times (SSRTs) and go misses.**

| Variables | Vocalization $M \pm$ SE [95% CI] | Finger tapping $M \pm$ SE [95% CI] |
|---|---|---|
| Go-asynchrony$_{\text{relevant stop}}$ | −35.37 ± 7.16 [−50.01 to −20.74] | −70.24 ± 8.62 [−87.86 to −52.62] |
| Go-asynchrony$_{\text{irrelevant stop}}$ | −61.47 ± 5.92 [−73.57 to −49.37] | −128.29 ± 9.19 [−147.07 to −109.51] |
| SSD$_{\text{relevant stop}}$ | 207.15 ± 8.01 [190.75–223.54] | 176.82 ± 10.81 [154.70–198.93] |
| SSRT$_{\text{relevant stop}}$ | 187.81 ± 9.52 [170.74–205.56] | 152.95 ± 6.60 [140.46–156.60] |
| Go misses$_{\text{relevant stop}}$◆ | 1% | 3% |
| Go misses$_{\text{irelevant stop}}$◆ | 0.4% | 9% |

Note:
   M, mean in milliseconds; SE, Standard error; 95% CI, 95% confidence intervals;
   ◆ This is the percentage of misses.

## APPENDIX I. CHECKING ASSUMPTIONS OF THE STOP SIGNAL TASK

The stop signal task is based on a model called the horse race model developed by *Logan & Cowan (1984)*. This model assumes independence of go and stop processes because failed stop RTs occur earlier than go-RTs. A finding that suggests in a failed stop trial, a response is given because the go process won the race against the stop process. In our current study, we did not have RTs but go asynchronies. We assume that in a failed stop trial, a response is given if the go process won the race against the stop; this would mean that the failed stop asynchrony would occur earlier compared to the go asynchrony. To check this assumption, we carried out a 2 × 2 repeated measures ANOVA with the within-subject factors of 2 go categories (go asynchrony$_{\text{relevant stop}}$, failed stop asynchrony$_{\text{relevant stop}}$) and two response modalities (finger tapping and vocalization).

The results show that the main effects of both factors were statistically significant but not the interaction between these two factors. The go category factor

($F(1, 29) = 62.20$, $p < 0.001$, $\eta_p^2 = 0.69$) revealed that indeed the failed stop asynchrony$_{\text{relevant stop}}$ ($M = -89.97$, SE = 6.84, 95% CI = [−103.94 to −75.99]) occurred significantly earlier compared to go asynchrony$_{\text{relevant stop}}$ ($M = -52.81$, SE = 7.28, 95% CI = [−67.69 to −37.92]). A finding that meets the horse race model assumption, suggesting the go process won the race against the stop process in the failed stop SR.

Moreover, the response modality factor ($F(1, 29) = 52.12$, $p < 0.001$, $\eta_p^2 = 0.65$) showed that asynchronies of finger tapping ($M = -91.76$, SE = 7.78, 95% CI = [−107.66 to −75.86]) occurred significantly earlier compared to those asynchronies of vocalization ($M = -51.01$, SE = 6.64, 95% CI = [−64.59 to −37.44]).

## ACKNOWLEDGEMENTS

We thank Prabhanjali Peters and Jessica Crampton for helping out with data collection. We also thank Jordan Wehrman for his valuable feedback on first drafts of this manuscript.

### Funding

Leidy J. Castro-Meneses was supported by Macquarie University Research Excellence Scholarships (MQRES) and the Perception in Action Research Centre (Post-doctoral Research Fellowship). Paul F. Sowman was supported by the National Health and Medical Research Council, Australia (#1003760) and the Australian Research Council (DE130100868). The funders had no role in study design, data collection and analysis, decision to publish, or preparation of the manuscript.

### Grant Disclosures

The following grant information was disclosed by the authors:
Macquarie University Research Excellence Scholarships (MQRES).
Perception in Action Research Centre.
National Health and Medical Research Council Australia: #1003760.
Australian Research Council: DE130100868.

### Competing Interests

The authors declare that they have no competing interests.

### Author Contributions

- Leidy J. Castro-Meneses conceived and designed the experiments, performed the experiments, analyzed the data, contributed reagents/materials/analysis tools, prepared figures and/or tables, authored or reviewed drafts of the paper, approved the final draft.
- Paul F. Sowman conceived and designed the experiments, performed the experiments, contributed reagents/materials/analysis tools, authored or reviewed drafts of the paper, approved the final draft.

## Human Ethics

The following information was supplied relating to ethical approvals (i.e., approving body and any reference numbers):

The human research ethics committee at Macquarie University granted ethical approval to carry out the study within its facilities (5201200035).

## Data Availability

The raw data have been supplied as a Supplemental File.

## Supplemental Information

Supplemental information for this article can be found online at http://dx.doi.org/10.7717/peerj.5242#supplemental-information.

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
