# Peer review of "Stop signals delay synchrony more for finger tapping than vocalization: a dual modality study of rhythmic synchronization in the stop signal task"

_PeerJ, doi:10.7717/peerj.5242_

## Round 0.1 · original submission · Major Revisions

Although the study protocol is interesting, the reviewers have raised several concerns regarding the interpretation and analyses. In particular, the study needs to be better integrated existing research and theory. The reviewers also raised concerns about the timing accuracy of the data and provided many suggestions to improve the writing. Finally, alternative explanations of the presented results should be considered.

·

Basic reporting

Language could be considerably improved by checking spelling and grammar, and by making the sentence structure clearer. More linear definitions of terms and dependent variables would advance the article and increase readability.

A great deal of sensorimotor synchronization literature is missing, including shadowing tasks that use vocalisations. Moreover, the theories on multimodal integration are not discussed and, as such, results are not discussed through these theories. Theories on error correction would also be useful for interpreting results.

The stop signal task should be described in full before the method section. Discussing it in the method section means that the assumptions and meaning of this original test are ignored and the focus is on the task. This background is necessary to understand the rationale for using this task and interpreting the present results.

In many sections, APA formatting is not adhered to, often at the cost of scientific writing or information that would aid the reader in interpreting the results. Figures could be improved by including all of the results instead of the ones the authors believe are important.

Hypotheses are not supported by previous results or existing theory. Having a larger discussion about the theory will strengthen the manuscript.

Experimental design

The research question in interesting but it needs to be framed in theory and existing research. The investigation suffers some methodological and technical flaws.

The method and measurements are unclear and not well-defined. More detail is needed and the steps need to have a linear description. The task is somewhat clear but how the measures are derived from the responses (or what the measures represent) requires more description.

Presentation software has a delay when interpreting button presses and vocal responses, and these are different between the two. Details for how this was calibrated are necessary. You would likely need to adjust the timing for the two measures. Also, how were finger responses recorded? How can you be sure that the two are equivalent for measuring the NMA? What was the vocal threshold? This certainly could have affected the detection of onsets. Since the tapping task gave auditory feedback, the literature on delayed auditory feedback is relevant here. Benchmarking of these tools and systems are necessary to draw distinctions between the two response types.

The description of the procedure could be simplified by referring to the manual task as “finger tapping” and the vocal response as “vocalising”.

The rationale for using a vowel sound is unclear. A hard consonant would have made onsets easier to detect and reach threshold faster. This could be problematic for the interpretation and comparisons to the tapping task.

The tasks are currently very difficult to understand and interpret. The figures do not help in clarifying the task. Also, the stop signal delay might be affected by the period due to entrainment. How many periods (beat inter-onset intervals) were examined? Was it only 400ms? This is missing information (at least early on) and could have been included in a “stimuli” section.

Validity of the findings

The discussion focuses on the positive findings and does ignores inconclusive results. Results were not discussed through theory and previous research is not explored.

Why were the other significant results presented? Why was “go type factor” removed? The results need to be reported in full for transparency. APA also requires p values greater than .001 to be reported in full. If there are effects that are uninteresting or unwanted (e.g., phase), then a linear mixed-effects model could be used to remove these effects. As it stands, the three-way interaction with go type, block type, and response modality needs to be explored.

Analyses are not explained. Justification or rationale for each analysis is not provided and it is unclear which hypotheses are being tested and when. For example, why does it matter that the first go in phase 1 was significantly different to that of phase 2? Why was it then removed? Moreover, other differences not reported. The authors could test if one ANOVA is a better fit than the other by comparing the two models. The analysis is currently unsound and unjustified. The mean of all of the stop signals should be used. Also, there are tests for statistical equivalence that should be used when the null hypothesis is accepted (e.g., Bayes Factor).

Figure 3 needs to show all conditions next to each other. Too much is hidden with the current arrangement.

Additional comments

The manuscript reports a good idea but the technical flaws need to be overcome first. Benchmarking the setup will help to mitigate concerns regarding latencies and delayed feedback, or delayed recording of vocal onsets. Moreover, a second experiment with a hard consonant vocalisation would make the results more convincing. I think that, should these changes be implemented, the paper could make a good contribution. The authors may also want to reconsider using Presentation as there are several latency issues for recording responses and producing auditory feedback using this software.

Specific comments

“Sensory motor synchronization” is usually called “sensorimotor synchronization”. In the present paper it is spelled both ways. Perhaps only use the latter.
Terms need to be defined “reafferent” and the notions of efference copies should be introduced and defined.
“I.e.” and “e.g.” should be followed by a comma throughout

Page numbers would be useful

Abstract
What is meant by “constraining the beat”?
Single and dual channel conditions need to be better defined. What does “manual” mean here?
Stop signals also need to be defined.
“Perturbed” NMA should be “reduced” NMA. Perturbed could mean increase or decrease depending on how it is viewed.

Check English and spelling, e.g., “… participant’s tap…” (line 56)

Missing definitions for some terms (e.g., “beat”)

Incorrect use of “which” (e.g., line 62). Replaced with “that” where necessary.

Line 64 “has to do with” sounds clumsy. Why not “is”?

Line 65. Should be “how regular taps are relative to the pacing stimulus”.

Line 74-75: Should be “differences… give rise”

Missing reference for “humans seem to perceive synchrony between the tap and the sound” (67-68). This would be an interesting paper. Same for “i.e. tapping to an auditory metronome is perceived to be 'in time', and not ahead of the beat.” (Lines 76-77) The perception of “in time” is not known, only the behaviour.

Lines 83-84: “cross sensory integration” should be “cross modal integration” to keep in line with the literature.

“Studies by Müller et al. (2008)” – only one reference, so should be singular, or should be more than one reference.

Lines 97-98: “Therefore, no central sensory multi-modal integration is required; hence, the possibility of delays being introduced by the integration process are minimised.”
Multimodal integration is required – proprioception, audition, and even touch (depending on the vocalisation type) are integrated. Unless you anaesthetise the mouth, you will still have multimodal integration.

Line 110: “cross modality interactions” should be “cross modal interactions”.

Line 111: “SR” used before it is defined.

Lines 112-114: “In the single channel SMS case, any shift away from the perception of synchrony is likely to be more apparent, and therefore more strongly resisted”
What is this based on? Why is it the “perception” of synchrony? This was not measured.

The stop signal task needs to be defined.

Line 130: +/- refers to a range, not the standard deviation.

Lines 132-133: Why was this range used? The rationale is not provided.

Line 137: Ethics number?

The “stop signal” is defined too late. The original stop signal task should be described in the introduction, including the rationale for why it is used and what it originally measured. The present method could then be reported more linearly.

By line 186 there are too many acronyms used and the acronyms are too similar. Perhaps shorten “Stop-signal” to “SS” then use the whole word following SS.

The purpose of the irrelevant stop blocks is unclear. Do they serve as a baseline?

Lines 203-204: “…measured as the increment in RT in the presence of intermittent stop signals”
Which increment? Is there a word missing? Do you mean the asynchrony for stopping/not stopping at approximately chance level?

Line 204: “In analogy with…” should be “analogous to”

Lines 208-211: “as the difference of synchronisation response from the mean of the relevant stop block against the mean of the irrelevant stop block”
The precise dependent variables should be defined. This could also be shown in the figure. “The amount of slowing” is difficult to link to a precise value or measure.

Lines 211-212: “synchronisation response data is statistically equivalent” – should be “was”. Check tense throughout (lines 214-215). Also, there are tests for statistical equivalence that should be used here.

Line 213: Incorrect use of semicolons (and this occurred earlier too). Check grammar and avoid usage if correct usage is unclear.

Lines 202 and 219: Proactive and reactive inhibition are not defined except by how they are measured. What do they represent?

Lines 231-236: The measure here is unclear. Why was the number chosen this way? What is the rationale and advantage of this method?

Line 239: APA requires numbers less than 10 to be written out. Use of hyphens unnecessary here.

Lines 243-250: This sounds like conditions that were counterbalanced. Order effects should be examined to account for learning and/or fatigue.

Line 252: Figure 1 does not show the instruction screen

Figure text is generally too small. Also, APA format is not used (e.g., Figure in italics, period after figure number)

Lines 287-289: Why were these SR values chosen? What was the rationale? What is the a priori reason that the first and second go trials are different from the others?

What was important about phase 1 and 2 that required them to be analysed? Why wasn’t order include as well (i.e., the pseudo-randomisation)?

Why were the other significant results presented? Why was “go type factor” removed? The results need to be reported in full for transparency. APA also requires p values greater than .001 to be reported in full. If there are effects that are uninteresting or unwanted (e.g., phase), then a linear mixed-effects model could be used to remove these effects. As it stands, the three-way interaction with go type, block type, and response modality needs to be explored.

Analyses are not explained. Why are the analyses being performed? Which hypotheses are being tested? For example, why does it matter that the first go in phase 1 was significantly different to that of phase 2? Why was it then removed? Why are the other differences not reported? Why not test if one ANOVA is a better fit than the other by comparing the two models? The analysis is currently unsound and unjustified. The mean of all of the stop signals should be used.

Figure 3 needs to show all conditions next to each other. Too much is hidden with the current arrangement.

Lines 334-340: A 2x2 ANOVA of relevant/irrelevant and modality (manual/vocal) is necessary. Was there a significant difference between relevant and irrelevant conditions, or an interaction? This is on lines 345-347 but they reveal no interaction, so the analysis was not warranted.

Line 358: “…that the SSRTs were not statistically different” – should be “were not significantly different”.

Results are not interpreted in the discussion in light of the evidence and existing theory.

Lines 369-370: “In order that the central matching of these two sources of information occurs with overlap in the brain” – this is a vague statement. Where and how do you propose this happens? Missing reference.

Lines 375-376: “that the sensory integration that occurs between senses” – redundant “that occurs between senses”.

376-377: Unclear what is meant by “a broader range of tap timings”?

377: “assymetrical” – spelling

379-380: “However, the lack of any discernible difference in the measures of variance for tap vs. vocalisation times doesn’t support this” – scientific writing does not contain contractions.

405-406: “Furthermore, there is a significant overlap in the neural substrates that support inhibitory control and the production of rhythmic tapping tasks.” – Missing references. Which neural substrates and which tasks?

407: “show that top down influences onto motor cortex” – “from” the motor cortex?

·

Basic reporting

-The use of the term “channel” early on was somewhat confusing. My understanding from reading the full paper is that this refers to some aspect of a sensory modality, but it would help to include a clearer definition up front (i.e., in the initial Background section).

-There is mention of alternative theories to that of afferent conduction time in accounting for NMA/differences in NMA depending on the effector synchronizing with a stimulus. It would be nice to see a short overview of what these alternatives include.

-It would be nice to see some references for the point made in lines 112-115, about how single channel SMS is more robust than alternatives, or at least a qualification of this statement considering other factors. Findings demonstrating superior synchronization to auditory stimuli as compared to visual stimuli during SMS (across contexts in which visual feedback about finger taps is available) could be seen as counter to this idea. (e.g., Iversen, J. R., Patel, A. D., Nicodemus, B., & Emmorey, K. (2015). Synchronization to auditory and visual rhythms in hearing and deaf individuals. Cognition, 134, 232-244.)

Ease of readability:
-Typo on line 56: “participants”

-Having fewer acronyms defined within the paper (not including those commonly used across work in this field), can become challenging. Unless there are constraints on space I might prefer to have fewer acronyms.

-It seems as though number the figure sections in order based on how the content is addressed in the paper would make for smoother reading as well.

-In the caption for Figure 1, A there is redundant language in the second to last sentence.

-Line 351 includes an incomplete sentence, “A finding that meets the horse race model assumption.”

-Line 354 should include “on average”.

-Something weird happens with the parentheses used in lines 408-410.

Experimental design

-The authors claim that by employing a vocal synchronization task the “dominant feedback modality” is of the same sensory channel as the stimulus and that they are “eliminating transduction and conduction discrepancies”. Based on the current explanations provided I am not convinced that it was necessary to adopt this new task in order to achieve a feedback modality consistent with the stimulus. In other words, wouldn’t adding an auditory feedback signal associated with button presses also accomplish this? There are other factors worth considering here as well such as the fact that participants have visual information about their button press movements, but not their vocalizations. The interesting question here seems to be about the effect of congruency between the sensory modality associated with the stimulus and the outcome of synchronizing behavior. If this is the case, it seems as though some alternative designs may have provided for better control. Additionally, if auditory feedback was produced in both the manual and vocal response conditions and the same pattern of differences in NMA were observed this might support an alternative explanation of the observation of differences in NMA for different sensory-motor synchronization tasks, for example one based on the time required to execute the required motor response or the kind of timing process supporting SMS (see e.g., Studenka, B. E., Zelaznik, H. N., & Balasubramaniam, R. (2012). The distinction between tapping and circle drawing with and without tactile feedback: an examination of the sources of timing variance. The Quarterly Journal of Experimental Psychology, 65(6), 1086-1100. for discussion of some of these ideas.) (Iversen, J. R., & Balasubramaniam, R. (2016). Synchronization and temporal processing. Current Opinion in Behavioral Sciences, 8, 175-180. also provides discussion of the idea that timing processes during SMS reflect perceptual goals.)

-As clarification in the Apparatus section, what was the monitor being used for? (Last sentence.) Also, did the button press make sound at all?

-It would be helpful to have a rationale or some other examples from SMS studies relating to the selection of a stimulus onset asynchrony interval of 1250 ms. This seems to be on the longer side, as previous work has determined that beats are most strongly felt when the interval is ~700 ms, and above 1800 ms synchronization is no longer possible (see e.g., London, J. (2002). Cognitive constraints on metric systems: Some observations and hypotheses. Music Perception: An Interdisciplinary Journal, 19(4), 529-550.)

-The description of participant instructions would benefit from being more explicit, specifically with regard to what they were told about the relevant stop blocks.

-In the section on Estimating proactive inhibition, I think I understand correctly that only “go” trials are being analyzed. This could also be more explicit for ease of understanding. The later discussion of how stop blocks may be influencing behavioral control is good, but having this be clear when the idea of proactive inhibition is first introduced would be helpful.

-Similarly, in the section Effect of stop signals on the synchronization response it would be helpful to state that this measure uses just trials in which participants didn’t successfully stop.

Validity of the findings

-It seems important to address the fact that vocal responses are occurring after the pacing signal, on average. This is true in the irrelevant stop condition and then in the stop condition they come even later. It’s possible that this means the threshold participants are using as the onset of vocalization is just much lower than the experimental threshold being employed, but if that’s the likely explanation it should be included in the Discussion. As is, it does not follow that vocalization timing is based on achieving simultaneous arrival of self-feedback and the pacing signal. This is also an important consideration with respect to discussion of why there is more reduction in the manual response NMA than the vocal response NMA. Given that vocal responses are already occurring after the pacing signal in the irrelevant stop condition, it’s possible that they can not come the same amount later as seen for manual responses in the relevant stop condition without interfering with the processes involved in the subsequent synchronization response (i.e., is there a possibility of a ~ceiling effect for reduction of NMA).

-The implication of the lack of difference in SSRTs between the modalities is not obvious to me. It would be worth including a mention of this in the Discussion as well.

Additional comments

Overall this study appears to be well-executed, particularly in terms of the analyses selected and conducted. The majority of my comments relate to areas where additional explanation or clarifying language would be helpful for reader understanding. However, I do have some reservations about how the experimental design addresses the questions of interest as they are currently articulated. I think the study is interesting and significant as conducted, but that it would be even stronger with additional explanation addressing the questions I've included with respect to experimental design (also possibly including the conceptual rationale provided in the Introduction).

·

Basic reporting

The study adresses the mechanisms underlying the phenomenon of negative mean asynchrony in (rhythmic) synchronization to a sound. To this end two tasks are studied and compared, tapping and vocal response (basically saying "i"). The hypothesis relates to the so called Paillard- Fraisse hypothesis, that the asycnrhony is caused by the phase shifting required to re-synchronize sound input and proprioceptive afferences. It is argued that in vocal response there is not such bi-modal integration mechanism, being a single sensory "channel": auditory. A stop paradigm, developed for the study of inhibition, is used to adress the robustness of the NMA (Logan et al.). The rationale is as follow: a single channel will be robust, a dual channel will be less robust to the "stop processing". In this case of rhythmic synchronization, the stop paradigm is adapted, basically a stop signal (cross turning to red) instructs the subject to stop synchronizing in the middle of a sequence. Irrelevant (don't stop) and relevant (stop) Stop signals are used.

I have two major comments: 1) the way the study is introduced focus on the NMA and the sensory uni vs bi modality distinction, while the Stop task is then presented as relevant to attentional and inhibitory processes. The prediction of a modulation of the NMA in each task is not presented initially as an attentional effect, the hypothesis tested being that the integration of two delays is the cause of NMA. Somewhat the MS changes topics on the way. 2) The assumption that vocal response is a single channel, auditory, is far stretched. The role of proprioceptive sensory mechanisms are known to be important in vocal function. Moreover as far as tapping is concerned, if a sound, a "clic", is added to the response, and the task is to sync the clic with the external stimuli, then vocal and tapping responses seem very similar. such tapping tasks have been studied in the NMA context. I would add also that there are data on the mean asynchrony for tapping in sync with a tactile stimuli, tapping in the aire without physical contact, and tapping at different rate which affects NMA.
The prediction of a more "tolerant" (from the authors) single channel process (vocal response) causing a greater resistance to the inhibitory stop mechanisms should be clearly delineated, compared to the assumed dual-channel tapping processes.
For an illustration, the role of the validity of the horse race model ine the Results section is not introduced earlier.

This said, it doesn't mean I don't appreciate the interest of the paradigm and data, but the MS should be thought differently to my opinion to make it informative and logic.

Finally, I would regreat that 30 years of research on synchronization to sound of tactile stimuli are completely ignored (Kelso et al., 1990). This has consequences in terms of theory first, but also in terms of observables founded on the theory. Of course the concepts are distincts, but there are also some basic emprirical results and methods that could be useful even if one doesn't adopt a self- organized, dynamical, account of cognitive and sensorimotor functions. In particular the stability of synchronization is a very robust observable (estimated by the variance (or std) of synchronization "errors").

Experimental design

Below some minor comments.

1) which device was used to record the vocal responses ? I guess the sound card.
2) the stop paradigm is described in great detail but could be introduced in a first simple and more transparent way, the reading is tenious to understand what the subjects were doing.
3) 161 : typo "go trials"
4) 211-221 : "The synchronization...equivalent to.." : I don't understand. It is the same variable or not? What means "statistically equivalent"?
5) 215-216: What means "we are presenting a synchronization response..."?
6) in Figure 3, one variable is "Proactive inhibition", this does not appear in the methods of the 1.3.3. stop signal sync task.
7) in the Results: 335 a typo: relevant

Validity of the findings

The data are sound, as the experimental paradigm. the discussion of "the idea that inhibitory control and rhythmic timing are linked" is interesting. The proof from the comparison of the two tasks is left to the reader, as no formal framework, non ambiguous and scientifically coherent to compare tasks does not exist to the best of my knowledge. Here is put forward the idea that one task is more single channel than the other, among other things.

But the logic of the MS, as stated above, is ackward.

Additional comments

I have nothing to add to my comments above

---

## Round 0.2 · Major Revisions

While the reviewers indicate the revised manuscript has substantially improved, they've also identified some outstanding issues that need to be addressed, in particular regarding the experimental setup and the timing of the stimuli and responses.

·

Basic reporting

Improved but Figures need to be formatted appropriately and null results should not be discussed.

Experimental design

The main methodological flaw (i.e., differences between veridical and measured responses were not obtained) is still problematic. See comments for a full description.

Validity of the findings

Difficult to assess until the correct calibration occurs. The null result is still interpreted as a "finding" and several interactions are not examined or controlled for using the correct statistical procedure (e.g., if "phase" is not important, then use a linear mixed-effect model to ensure it does not moderate the effects).

Additional comments

General comments
The authors have improved the manuscript considerably and several aspects are much clearer. However, the methodological concerns regarding differences between the data collection method of the voice controller and button response remain. I understand that the authors have attempted to address this issue so I describe what is necessary in full here. When a program (e.g., Presentation) records the timing of a stimulus, there is often some delay (I believe approximately 4ms after the time it reports for presentation). However, when presentation has to make auditory feedback relative to a response (e.g., keypress), this delay is much larger (I have measured approximately 40ms +/-5ms); some of this time is the error of recording the veridical response and the rest is error in producing the auditory feedback. To calibrate the system correctly, one needs to measure the veridical onset of the response (finger tap or vocalisation) relative to the stimulus. Currently, what is shown is the detected vocalisation onset relative to the start of audio recording by Presentation but this does not necessarily reflect when the stimulus occurred and this only lets one know what Presentation recorded as opposed to what actually occurred. One method for obtaining the veridical timing of the finger taps and vocalisations relative to the stimulus is to use any oscilloscope (or analog input box or audio sound card) that records the signals for all devices simultaneously using different channels (see Schultz & van Vugt, 2015 for a simple example). By the “signals” I mean both the inputs and the outputs; if there would be auditory feedback (although, in this case there is not), then one would need to record the veridical timing of the keypress through some kind of sensor (e.g., force sensitive resistor) and the auditory output (I mention this just in case the authors reasoned that using auditory feedback of keypresses is a viable measure of veridical onsets; they are not). What is missing for the present experiment is a calibration that measures the continuous signals of the stimulus, keypresses (through a sensor), and speech audio to obtain the veridical latencies of responses and to compare these to those recorded by Presentation to see if any correction needs to be applied. If the latencies are constant within a given range (e.g., any value with a range less than +/- 0.5ms), then you might be able to subtract these baseline values from each condition. If they are more variable, then the experiment might need to be replicated with the veridical latencies recorded in the manner described above. As it stands, the system has not been adequately calibrated to make any claims regarding differences between finger taps and vocalisations. Once this issue has been resolved, then the main claims would be justified. Until then, no claims can be made regarding negative mean asynchrony (i.e., hypothesis 1), although the claim regarding the change within response conditions (finger, vocalising) still stands (i.e., hypothesis 2) as each condition is relative to a baseline (i.e., tap irrelevant minus tap relevant, vocalisation irrelevant minus vocalisation relevant).

There are also references on delayed feedback and modality (e.g., Sugano et al., 2012, and others on tactile feedback) that are missing and should be included.

Last, differentiations between “in the brain” and “in the world” should be termed “perceptual” and “veridical” (or similar). “In the brain” would require EEG or fMRI.

Minor points
Title
Don’t use double negatives, suggest change to
“Stop signals increase synchrony more for finger tapping than synchronized vocalization: a dual modality study of rhythmic synchronization in the stop signal task”

Also, “synchronized vocalization” assumes it is “synchronized”. Suggest removing “synchronized”.

Abstract
“Paillard-Frassie hypothesis” should be “Paillard-Fraisse”

Vocalisations are still referred to as “single modality”. There is tactile feedback there too. What about bone conductance? Also, keypresses tend to produce auditory “clicks” that technically *could* be auditory feedback. These limitations won’t stop the paper from being published but should be acknowledged.
“Less asynchronous” would require the absolute value to be tested (i.e., the magnitude of asynchrony).

Bayes factor tests are necessary if you want to interpret the null result between finger tapping in the synchronization and stop-signal task. It is likely to give evidence for the null hypothesis given the means but the test should be performed.

Line 82
“e.g.” occurs without a comma. This was mentioned in the previous review and authors claimed to have fixed it. This also occurs for “i.e.” (e.g., line 113).

Lines 130-131
Specify that the t-tests showed “no [significant] differences”

Lines 143-146
This description is confusing as it makes it sound like there is auditory feedback for the tapping condition. Perhaps first state that the pacing signal involves the auditory modality, finger tapping adds the tactile modality, and vocalising adds the tactile and a second auditory modality.

Line 165
“On go trials” should be “”In go trials”

Line 177
“E.g.” should be inside parentheses

Line 202
No capital should occur after the semicolon.

Lines 209-212
Hypothesis should be statements (e.g., x is y), not possible future suggestions (e.g., x would be y).

Line 205 (and elsewhere)
Remove the “registered” logo after Presentation. I believe this is not necessary.

Lines 250-254
What was the hardware of the PC? Some PCs cannot produce 120 frames per second and the “syncmaster” does not work unless you have a compatible graphics processor unit.

Lines 288-291
I know it’s picky but with 120Hz, your visual timing resolution is 8.3ms. This means that, technically, your increments would only be approximately 30ms (to the nearest multiple of 8.3ms).

296-297
“Inter-tap interval” should be “inter-onset interval”. The former is the measure whereas the latter is a property of the stimulus.

324-326
The description and term used for “synchronized response” are still unclear. Why not “response time difference”? Also, shouldn’t this measure the difference between asynchronies instead? Why is the response time difference for continuous responses important? Why not call them “asynchrony with pacing signal” and “asynchrony between tasks/relevance”? I still don’t know why the latter is important, but it would make the terminology clearer.
How were missed responses dealt with? Wouldn’t this increase latency if a cycle was missed?

351
Separate Appendices should be used for each section and should be labelled as they appear in the manuscript. Also, the figures and tables in the appendices should use a difference nomenclature than those in the manuscript (e.g., Figure A1)

372
“…in time to…” should be “…in time with…”
Line 386 (and throughout)
Figures need to use APA formatting and should not have borders. “Figure n.” should be italicised and the caption should continue on the same line.

406-414
Why only these two? What about the other events? Wouldn’t that make a better baseline? From the table on lines 446-447 it appears that there is an interesting difference between “all other go” where finger tapping has an advantage.

413-414
“We were interested on…” should be “We were interested in…”

416-420 (and throughout results)
If p is less than .001 then one can report “p < .001”. Currently it says “p = .001”. Is this the case or was it less than .001?

430
Should it say (ps > .41)? Was it one p or multiple? If one, then report one p value (p = .41). If multiple, then say what value the ps were greater than.

440-441
If all other pairwise comparisons were significant then there was a significant difference between the relevant and irrelevant stimuli for the vocalisation condition: 5.92ms vs. 5.96ms. I don’t think this is the case, and the results need to be reported more precisely.

468-473
“…occurred significantly closer to the auditory pacing signal compared to the asynchronies…”
To say this (and the later statements), the absolute asynchronies need to be used. This becomes more confusing in lines 473 to 478 where the magnitude of the vocalisation (32.06ms) is reported as “similarly” closer to the pacing signal than the finger tapping (5.59ms).

489-490
“did not differ from the onset of the pacing signal” – See previous comments regarding “significant differences”.

Figure 3 is still not transparent. All of the factors and associated levels should be shown (see previous comments).

532
“Paillard-Frassie” – Misspelled “Fraisse” and missing “hypothesis”. The plural “hypotheses” at the end is awkward since these are the names of the hypotheses (e.g., “the Paillard-Fraisse hypothesis” is the name).

594-595
“made finger tapping very accurate” – should be “made finger tapping more synchronous”.

599-603
Where are the references for these studies? The reference you discuss is a review paper but I could not find the “hold back” quotation used here.

597-599
“did not find differences” – should be “did not yield a significant difference”. Check throughout the article for these mistakes.

632-633
“An incidental finding worth highlighting in this study is that SSRTs were not different across the vocalizations and finger tapping.”
A null result is not a finding and definitely not one worth highlighting. A null result is likely due to variability/noise in the experimental setup, or it indicates that you need more participants to find an effect. Bayes factor tests are used to measure whether one can accept the null hypothesis and the strength of the evidence. As this test was not used, the conclusions that are in the paragraph (632-647) are void.

828
The significant (and near-significant) interactions with phase require exploration. What do these interactions mean? Table 3 does not help to clarify the issue and requires the descriptive statistics of all factors used in the ANOVA.


References
Sugano, Y., Keetels, M., & Vroomen, J. (2012). The build-up and transfer of sensorimotor temporal recalibration measured via a synchronization task. Frontiers in psychology, 3.

·

Basic reporting

I initially commented “There is mention of alternative theories to that of afferent conduction time in accounting for NMA/differences in NMA depending on the effector synchronizing with a stimulus. It would be nice to see a short overview of what these alternatives include.” The authors responded with inclusion of additional theory relative to inhibitory control and SMS, but I feel the authors still need to explicitly acknowledge the many other alternative theories put forth to account for NMA as are outlined in Repp (2005) and Repp and Su (2013) (even if the mechanism is truly multi-factorial, as the authors suggest). This overview would be fitting within the discussion raised starting in line 91, before thorough discussion the Paillard-Fraisse hypothesis and its relation to the current study.

I suggested the inclusion of some additional references, which the authors do not see as particularly relevant to the scope of the current project. I maintain that the area of research that employs to the metric of performance variability for SMS (and that including studies that use other tasks for investigating sensorimotor performance) is important to the larger considerations of perception, timing and sensorimotor integration being addressed by the current project. As such I think some inclusion of key findings from these studies would help in situating the current study as relevant within this broader field.

Line 203 should contain “inter-modal”.

Experimental design

I initially commented ”The authors claim that by employing a vocal synchronization task the “dominant feedback modality” is of the same sensory channel as the stimulus and that they are “eliminating transduction and conduction discrepancies”. Based on the current explanations provided I am not convinced that it was necessary to adopt this new task in order to achieve a feedback modality consistent with the stimulus. In other words, wouldn’t adding an auditory feedback signal associated with button presses also accomplish this?” The authors have cited a study that does this, but do not include a description of the task in the paper and do not discuss the related findings. These seem critically important to understanding the background contributing to the current study.

I initially asked “Did the button press make sound at all?”. The authors responded with clarifying details that I think should also be included in the text of the paper, including the fact that participants were “encouraged to relax as much as they could and respond with ease, so this would not make a loud noise.” Furthermore, this leads to the question of what headphones were used, and were they noise-cancelling? These are all important details. In future uses of this paradigm it might also be good to survey participants afterward about whether they could hear their taps in order to be able to provide at least anecdotal information on this.

Relatedly, can the authors provide a rationale for selecting the vowel sound they did for vocal responses?

Validity of the findings

I initially commented “It’s possible that the threshold participants are using as the onset of vocalization is just much lower than the experimental threshold being employed, but if that’s the likely explanation it should be included in the Discussion.” This possibility is still not considered in the text, which seems problematic largely because a) the pre-set threshold for response detection is not specified in the manuscript (I think this should be included for replication purposes), b) there is no empirical or theoretical support provided for the selection of this threshold. This is the key remaining issue I see with the current manuscript as, based on my understanding, if the threshold was lower and caught vocalization onsets earlier, the difference in asynchrony magnitudes between synchronized vocalization and finger tapping may not actually be significantly different. If it were possible to reasonably demonstrate that the threshold selected corresponds to the point at which individuals perceive their vocal responses as coinciding with a periodic stimulus the findings would be much stronger.

The statement “asynchronies were reduced in the relevant stop blocks compared to the irrelevant stop blocks for the synchronized vocalization” (lines 575-576) is not consistent with the results as it is currently worded. While it may be accurate to say that negative synchronies were reduced, the synchronized vocalization asynchronies were not negative to begin with and actually became larger/appear to exhibit a shift to reduced accuracy in the context of the relevant stop blocks. This wording needs to be clarified throughout in order to avoid misstating the results, and there should be some discussion of why the stop signals appear to be making synchronized vocalizations less accurate (at least based on the chosen threshold for vocalization onset).

Additional comments

The resubmitted manuscript is much improved in all areas following revisions based on the initial review. It is now generally more coherent from start to finish and provides a much stronger rationale for the way in which the study was conducted, as well as a better background for appreciating the outcomes.

·

Basic reporting

The MS has been considerably revised and is aceptable for publication to my opinion in its current state.

I noted a typo in the abstract :
Typo ligne 38 The Paillard-Frassie hypothesis : Paillard Fraisse

Experimental design

no comment

Validity of the findings

no comment

Additional comments

The MS has been considerably revised and is aceptable for publication to my opinion in its current state.

I noted a typo in the abstract :
Typo ligne 38 The Paillard-Frassie hypothesis : Paillard Fraisse

---

## Round 0.3 · Major Revisions

The reviewers are still not fully satisfied with the revised manuscript. Both reviewers indicate that alternative theories should be discussed in more detail to better position the current work within the field. They also suggest to present the calibration in an Appendix. Results should be presented in full and several typos were identified.

·

Basic reporting

There are several grammatical and spelling errors remaining in the text

The literature review is still rather one-sided and the discussion does not refer to competing theories.

The figures are still not transparent and hide many of the results that could potentially be interesting to others in the field, or persons who would like to replicate the results.

Experimental design

The experimental design is well thought-out. The calibration definitely adds to the design and reveals different results. I am still worried that the chosen method of calibration adds extra noise because it relies on the voice key. The method described in "General comments to the author" avoids this issue and should be implemented. I understand this is annoying but I hope the authors understand given that the previous calibration already revealed vastly different results. This would then satisfy the criterion "Rigorous investigation performed to a high technical & ethical standard"

Moreover, the calibration process should be included in an appendix, including a description, results, and figures to conform to the criterion that "Methods [are] described with sufficient detail & information to replicate."

Validity of the findings

As stated below, the findings are valid but alternative theories should be discussed.

Additional comments

General comments
The authors have done a great job of responding to the comments, particularly the calibration of the measurement tools. I recommend the calibration process be put in an Appendix with the full results and figures to make this clearer to the reader. I also recommend the authors consider the more robust calibration method of recording the audio (stimulus, vocalisation) and voltages (button press) simultaneously as the current method confounds the variability and latency of the vocal onset detection itself. Given the amount of noise in the system, the authors may find that some results that were previously not significant (due to noise) could be significant. This is necessary for considering the alternative hypotheses that are currently not discussed.

The statistics are not reported in full and the full descriptive statistics and pairwise comparisons are not provided. These are necessary for full transparency and so the reader can also interpret the results (significant or otherwise). This is particularly important because the results could be interpreted through other theories that are not discussed here. By only presenting the results that support the current theory, support for other theories cannot easily be assessed.

Much of the spelling and grammar require proofreading and a more thorough literature review is missing (e.g., line 87 “Research suggest[s]”). More care should also be taken when reporting the statistics as there appear to be errors regarding directionality (missing negative symbols) and the values used (or, at least, the description of how the means have been derived).

The term “proactive inhibition” is not defined in the abstract and its use is vague. Why not just inhibition or vigilance? If this term has a specific meaning then this should be provided in the abstract before its use. This term has a particular meaning but it appears to relate more to memory interference and motor transfer between tasks than cue-based stopping. The reference for this term is an unpublished thesis (Castro-Meneses et al., 2015; note the year is incorrect in the reference list and text) and not a peer-reviewed text that defines “proactive inhibition” in this manner. Has anyone else used the term in this fashion? If not, the rationale for the use of this term (and reactive inhibition) should be more detailed. Given that responding in the irrelevant stop signal condition can be more automatic than responding in the relevant stop signal condition, results could reflect differences in vigilance, automaticity (in line with Repp and Su, 2013), or even attention and cognitive load (see Maes et al., 2015). These alternative theories should also be discussed.

Specific comments
Line 217
“Synchronicity” is used whereas “synchrony” is the correct term

Line 260
The audacity reference and website should go in the references section.

Line 451
Additional “fi” in F value

Line 482
These comparisons do not examine the three-way interaction – they only examine the main effect of stop block for all of the different conditions. You require all comparisons here to explore the interaction appropriately.

For results with negative values, do not use hyphens. Use “to” or a comma to separate the values (e.g., in CIs) as the minus symbol is sometimes missing (e.g., lines 501-502). Moreover, these statistics do not match up with those in Table 1.


Author responses to reviewer comments
> We have made a calibration procedure that yielded this null result as significant. Please see full
> comments next. In relation to the phase analysis, the used statistical analysis is valid because an
> ANOVA is a special case of linear mixed effect model (it is a linear model with only fixed effects).
An ANOVA is not a linear mixed-effects model because there are no mixed effects. The interactions must be examined and covariates should be accounted for if the authors choose to assert that some variables (e.g., phase) are not important.

> There are also references on delayed feedback and modality (e.g., Sugano et al.,
> 2012, and others on tactile feedback) that are missing and should be included
The authors missed the “and others…” – I gave an example reference but the literature review needs to be more complete.

Aschersleben & Prinz (1995) reference
The terms “in the brain” and “in the world” are outdated, even if Aschersleben and Prinz used them in 1995. Neuroscience has come a long way since then and the term “in the brain” means something functionally different than it did previously. If you choose to use these terms as descriptors, then link them to the processes or functions that represent them in the modern literature and only use the old nomenclature once (to make the link with Aschersleben & Prinz). Alternatively, discuss the neuroimaging literature to support invoking “in the brain” although I do not think this is the direction in which the authors were going.

> We like synchronized vocalizations, we have had long debates and come with that consensus
Unfortunately I was not part of those discussions and the manuscript now uses an incorrect term. Either use “vocalization” or find another term that does not assume synchrony from the outset.

Regarding the “Paillard-Fraisse” hypothesis
> Good pick up. Thanks. This has been corrected throughout the manuscript.
The spelling is still incorrect. You now write “Fraissie” in some sections (e.g., abstract)

Regarding hypotheses
Changing the hypotheses to “will” statements still uses future tense. It should be a statement of something that *is*. For example, one of your hypotheses should be “Vocalizations to auditory pacing signals are hypothesized to exhibit reduced negative asynchronies compared to finger responses”.

Figure 3
All the levels and factors need to be described or a table with the full descriptive statistics should be included. Again, this is a matter of transparency and all effects should be discussed so the reader can assess the results. Moreover, the lines between the figures are not APA format (also, legends should be inside the figure, Figure 3C requires annotations to show that the range does not start at zero, significance levels usually do not go in the figure). Figure 3A only shows a two-way interaction.

Aschersleben 2002 quotation
There are APA formatting guidelines for citing a citation and these should be followed. The colloquial use of ‘hold back’ should be described in more concrete terms.


Calibration method
Please include the calibration in an Appendix so the reader can assess how these tests were performed and what can be ascertained from them. The full results of this test should be included because there are several sources of latency and variability. The current description in the response could refer to several different aspects of the setup and it would be good to assure the reader of the test validity and show the results for:
1. Latency of auditory feedback following button press
2. Latency of recorded onset times for vocalisations relative to stimuli
3. Latency of recorded onset times button presses relative to stimuli

It would be easier if the full write-up were included because the brief description is difficult to parse without figures. I believe that the third calibration test has not been performed and the second test was performed using an indirect method. Both of these tests relied on the vocalisation recording and threshold. A better method would be to record the stimulus audio, the vocalisation audio, and voltage change from the accelerometer for the button response. This would show the timing difference between the recorded asynchronies and the veridical asynchronies. The current test is suboptimal as it confounds some latencies and variability (i.e., the vocal response measure is a weak point) but it is a good start.

On the topic of the button response, Accelerometers are not the best tool for measuring onsets as they contain inherent delays in ascertaining velocity and are triggered a while after the response has been made (much like the microphone, as the vibrations ramp). A force sensor (FSR) or pull-up is recommended instead as these provide a quicker and more accurate signal.

In your description, was the response device a “headphone” or “microphone”? It should be the latter, no?

References
Maes, P. J., Wanderley, M. M., & Palmer, C. (2015). The role of working memory in the temporal control of discrete and continuous movements. Experimental brain research, 233(1), 263-273.

·

Basic reporting

It seems as though the authors misinterpreted my suggestion to cite Repp (2005) and Repp and Su (2013). This was not meant to encourage a discussion of phase correction and period correction, but rather an overview of the alternative hypotheses put forth to explain NMA specifically. These papers provide a summary of the theories generated surrounding the NMA phenomenon, including the Paillard-Fraisse along with several others. This is still missing from the current manuscript.

Based on my understanding of the Paillard-Fraisse hypothesis it meant to explain NMA, but does not speak specifically to phase correction. I believe a discussion of phase correction vs. period correction is relevant to the current study, but the way it is currently incorporated is making things muddled.

The inclusion of references for work examining performance variability is helpful for situating the work within the broader relevant field As I noted in my previous review, it would also be nice to see a sentence or two describing how the outcomes of this work as well as work examining asynchronies have contributed to our current understanding of SMS. If you want to keep a mention of the theory on phase correction and period correction processes this would be a good place to do it.

Experimental design

In my last review, I asked the authors to provide details about the equipment used, the instructions to participants and the vocalization selected. This information in this revised manuscript is very helpful.

I suggested “in future uses of this paradigm it might also be good to survey participants afterward about whether they could hear their taps in order to be able to provide at least anecdotal information on this”. This was meant for the authors to keep in mind, I’m not sure mentioning it within the manuscript is necessary.

Validity of the findings

The authors have provided a measurement for the vocal onset threshold based on the settings for the software used, but no units corresponding to this measurement. Is this in decibels?

The additional information about showing participants how the presentation software detected their vocalizations is helpful and informative as is comparison of different methods for detecting vocal onset.

While I understand the general idea behind the calibration of responses it is somewhat unclear in the manuscript why this was unnecessary. What is PowerLab doing? How were button press and vocal response times being recorded before that they weren’t veridical? More information is needed here about this process.

---

## Round 0.4 · Major Revisions

The reviewer indicates that his concerns have not been adequately addressed. As this is the third round of reviews, I would like to give the authors a final chance to address these concerns. The main issues that need to be addressed are the calibration and the use of terms. Appendix A does not provide a direct calibration between the output (sound beat) and the input B (button press). This needs to be added. If this cannot be added, this should be clearly stated in the manuscript and be acknowledged as a limitation in the Discussion. We also agree with the reviewer that the terms “in the brain” and “in the world” are outdated and need to be removed. Especially as they are used in quotation marks in the manuscript, it is unclear what they mean. For example, it is unclear how a statement like “This finding shows that participants tried to recalibrate their taps to a subjective experience ‘in the brain’ of temporal coincidence.” can be tested or falsified. The authors need to address all the reviewer comments in a point-to-point reply.

The manuscript will not go back for re-review; the editorial team will ensure that the required changes were incorporated into the manuscript. If the Reviewer still see issues with the revised manuscript, we advise him to publicly share his concerns so discussions on alternative interpretations and theories become part of public scientific discourse. Although PeerJ only considers Research Articles and Literature Review Articles, we encourage commentaries to be submitted to PeerJ Preprints.

·

Basic reporting

Several main theories should be discussed:
Modality Appropriateness (Hove & Schwartze, 2014)
Dynamic Attending Theory (Large & Jones, 1999)
Multimodal/crossmodal integration (specfically those on tactile and auditory synchronization)
Phase and period correction (more than simply including the reference; see comments below for specific comments)

Again, the term "in the brain" is not appropriate here and no results are provided to show whether there is "subjectively perceived" coincidence or any coincidence "in the brain". There is only behaviour whereby participants could be integrating different types of responses and/or modalities.

Experimental design

There was no misunderstanding about the design of the calibration from my side. The authors did not correct calibrate their equipment and most tests relied on the voice key recording resulting in a "loop" (as opposed to a proper calibration). The voice key should only be used for the vocal condition. All conditions (including the vocal condition) requires audio to be simultaneously recorded with the stimulus (output) and the timings be compared with those provided by presentation. Here is a picture where the | represents the onsets of the audio over time.

Test for Keyboard:
----------------------------------------------------Time---------------------------------------------->
Stimulus Audio:-----------------------|----------|----------|----------|----------|----------|----------|----------|
Keyboard sensor/audio:-------------|-------|---------|---------|---------|---------|-----------|----------|-----


Test for Vocalisations:
----------------------------------------------------Time---------------------------------------------->
Stimulus Audio:-------------|----------|----------|----------|----------|----------|----------|----------|
Vocal audio:------------------|---------|-----------|--------|-----------|---------|-----------|----------|


The authors will measure the onsets for the responses (keyboard/vocal) relative to the stimulus. They then compare these times to those given by Presentation and report the differences and variability. This is a correct method to measure the veridical latency of the device(s) and is required given that the direction (positive/negative) and magnitude of the mean asynchrony is under investigation here.

Validity of the findings

As stated above, the correct calibration method is required. I will review the results section more thoroughly once this has been performed.

Interpretations regarding perception and anything "in the brain" should be identified as speculation and only occur in the discussion. The neural data that is currently reported is not linked to the current study and should either be removed or show how these previous findings can explain the present results.

Currently proactive inhibition is not linked to any of the SMS literature. This is strange because it shows a nice dissociation between controlled and automatic processes (such as those discussed by Repp and others; see Mills et al., 2015 or other work by Keller). Moreover, the Paillard-Fraisse hypothesis does not offer an explanation for this process so it seems odd that it is the favoured theory here.

Additional comments

The authors did not satisfactorily address the comments from the previous reviews. I provide detailed comments here but the main problems are the calibration process, the interpretation and chosen theory (which does not adequately explain the data in the current stop signal paradigm), and the discussion of neural data to support the interpretation of results. The authors now repeat the term "In the brain" despite the previous comments that 1) there is no evidence to support neural responses to coincidence, and 2) there is no perception task to show what is perceived as "coincident". Currently all of the results could be explained by different strategies when using different effectors for different tasks (matching the action with the stimulus compared to matching the produced sound with the stimulus), different physical constraints, or differences between the number of modalities used (i.e., vocal uses two, tapping uses one). It is likely a combination of all of these but there is no evidence that this is due to differences in perceived coincidence presented here so one cannot assess whether the tapping responses were actually perceived as different with regard to coincidence.

Specific comments (lines refer to tracked changes document):

Line 136
“ … finger tapping try to minimize…”
Should be “tried”. Also the “finger tapping” is not “trying” to do anything. It is the human who is trying to minimize energy usage during a finger tapping task.

Line 141
“…also Pikovsky, Rosenblum, and Kurths (2003) who gives a more holistic…”
Should be “that gives” – the reference is to the article not the authors.

Lines 149-150
The evidence is in favour of multiple theories. Dynamic attending is the main theory that is missed here.

Lines 155-158
Page numbers and citation missing for terms “in the brain” and “in the world”.

Line 160
“This unmatched timing coincide…”
This is not correct grammar and does not make sense.

Lines 160-167
Missing reference for this time difference. Also, specifically state which tactile/kinaesthetic information takes longer and what it takes than (visual? Auditory?)

Lines 175-179
“This finding shows that participants tried to recalibrate their taps to a subjective experience ‘in the brain’ of temporal coincidence.”
These were not neural imaging studies so there is no evidence for “in the brain” temporal coincidence. Also, their subjective experience is not involved here as participants are often unable to report when there is a temporal delay unless it reaches a certain amount. This is simply a perceptual phenomenon and these findings have been interpreted as matching the tap feedback to the audio of the pacing signal (instead of matching the tap itself to the pacing signal). This is “recalibration” and “habituation” – not “in the brain” temporal coincidence. Unless you have iEEG evidence then this is purely speculative and also misrepresents the interpretations of the original authors. This belongs in the discussion and outlook sections – not the introduction.

Lines 181-185
Poor grammar in this sentence. Also, why is this paragraph separate from others? Evidence for different tactile afferent conduction times between the foot and hand? This could simply be different physical systems with different constraints.

Lines 189-191
This sentence does not make sense and is a fragment. What was meant by “was not different from the onset of the tactile pacing signal”? Did the authors mean “significantly different” because -8ms is different than 0ms?

Line 193
“Significantly negative”? Should be “significantly less than zero”

Line 194
This “contention” has not been discussed. Also, evidence cannot support a “contention” – evidence can “resolve” a contention.

Lines 195-211
This rationale makes no sense and, again, requires iEEG to actually test. In every case, the subjective experience will be different from the objective world – that is why we have perception. An inter-/ intra-modal comparison does not change this fact. Not only that, the perceptual system itself has its own inherent temporal resolution and delay and humans have habituated to this delay to perceive synchrony (or lack thereof) so what happens in behavior might not reflect the temporal coincidence of events occurring in the brain. One can have a “perception” of temporal coincidence but this is neither temporal coincidence “in the brain” nor is it necessarily a “subjective experience” of coincidence. Moreover, the authors don’t test perception of temporal coincidence – they test synchronization with a pacing signal. A task is missing where participants state whether they were synchronous or not (or how synchronous they were). Therefore, this line of reasoning should be removed and replaced by the original authors’ interpretations or an alternative explanation that can be supported with evidence from the current experiment. Conjecture and speculation belongs in the discussion.

Lines 226-227
Incorrect tense

Line 231
State the author instead of using pronouns (“they”).

Lines 247-248
Incorrect tense (should be past tense).

Lines 248-249
Why “unlikely to be precise”?

Line 311
Remove “In the brain”

Line 323
“Synchronicity” is not the correct term here (as stated in the previous review)

Lines 787-788
Repetition of “sensory modalities for the”

Lines 848-858
This paragraph requires further clarification. Also, why didn’t the authors look at tap-variability in the present experiment? The data is there and this would make an interesting analysis considering it is a one-experiment paper.

Lines 851-855
Why was this moved here? It belongs in the introduction and should be expanded upon. The whole discussion about automatic and controlled processes in phase and period correction tie in very well with the results and “proactive inhibition” forces what is usually an automatic process to become more controlled.

Lines 884-886
“…musical training may require learning to delay the response so it can occur” – this is unclear and does not make sense given that musicians already show less negative mean asynchrony.

Lines 903-906
This paragraph seems out of place and is not explained with regard to the results of the study. The authors need to make the link between these studies and their own study especially to support any notion of “subjective coincidence in the brain”.

Line 931
“longer in the vocalizations relative” should be “longer for vocalizations”

Line 1180-1181
“…that have a playback delay…” should be “that has a playback delay”

---

## Round 0.5 · accepted · Accept

The editorial team has reviewed the revised manuscript and the authors have adequately addressed the main issues described in the previous decision letter. Although no direct calibration between the sound beat and the button press is provided this limitation is clearly stated in the Discussion. As such the revised manuscript meets the criteria for publication in PeerJ.

#